# ENERGY-BASED DIFFUSION LANGUAGE MODELS FOR TEXT GENERATION

**Minkai Xu**[1]***Tomas Geffner**[2]**, Karsten Kreis**[2]**, Weili Nie**[2]**, Yilun Xu**[2]**,
Jure Leskovec**[1]**, Stefano Ermon**[1]**, Arash Vahdat**[2]
[1]Stanford University [2]NVIDIA

## ABSTRACT

Despite remarkable progress in autoregressive language models, alternative generative paradigms beyond left-to-right generation are still being actively explored. Discrete diffusion models, with the capacity for parallel generation, have recently emerged as a promising alternative. Unfortunately, these models still underperform the autoregressive counterparts, with the performance gap increasing when reducing the number of sampling steps. Our analysis reveals that this degradation is a consequence of an imperfect approximation used by diffusion models. In this work, we propose Energy-based Diffusion Language Model (EDLM), an energy-based model operating at the full sequence level for each diffusion step, introduced to improve the underlying approximation used by diffusion models. More specifically, we introduce an EBM in a residual form, and show that its parameters can be obtained by leveraging a pretrained autoregressive model or by finetuning a bidirectional transformer via noise contrastive estimation. We also propose an efficient generation algorithm via parallel important sampling. Comprehensive experiments on language modeling benchmarks show that our model can consistently outperform state-of-the-art diffusion models by a significant margin, and approaches autoregressive models' perplexity. We further show that, without any generation performance drop, our framework offers a $1.3\times$ sampling speedup over existing diffusion models. Reproduced code is available at https://github.com/MinkaiXu/Energy-Diffusion-LLM.

## 1 INTRODUCTION

In recent years, autoregressive (AR) models have achieved remarkable advances in modern large language modeling (Vaswani et al., 2017; Wolf et al., 2020; Brown et al., 2020; Chowdhery et al., 2023; Touvron et al., 2023), showing impressive results in tasks such as expert-level coding (Roziere et al., 2023) and reasoning (Wei et al., 2022; Trinh et al., 2024). Despite the considerable progress achieved, the left-to-right generative paradigm exhibits several long-standing drawbacks. For example, training and sampling of autoregressive models require a fixed ordering of the sequence, constraining their generation flexibility (Hoogeboom et al., 2022; Shih et al., 2022). Furthermore, at test time, the generation is heavily conditioned on previous generations, which leads to accumulated error *a.k.a* exposure bias (Bengio et al., 2015). As a result, alternative generative paradigms beyond left-to-right generation are actively being explored.

Discrete diffusion models (DMs) have recently emerged as a promising competitor for discrete sequence generation (Austin et al., 2021; Dieleman et al., 2022; Gat et al., 2024). Unlike AR models, discrete DMs conduct generation by progressively decoding the full sequence in parallel starting from a fully masked sequence, offering great potential in bidirectional controllable generation and sampling acceleration. Most recently, great efforts have been invested in improving their performance by extending discrete diffusions to operate in continuous time (Campbell et al., 2022; 2024; Sahoo et al., 2024; Shi et al., 2024). Unfortunately, despite their promise, these models still underperform AR models by a large margin, resulting in limited practical usage. Developing principled and powerful discrete DMs remains an open problem.

In this work, we take a closer look at the existing discrete diffusion models and reveal a vital mismatch issue between the training and sampling distribution in its current form. Specifically, the

---

*Work done during an internship at NVIDIA. Corresponding author. ✉ minkai@cs.stanford.edu

existing discrete diffusion models aim to predict all missing tokens in parallel at each intermediate diffusion step, but the denoising joint distribution is simply parameterized as a product of token-wise independent distributions. As a result, intermediate denoising steps ignore sequence-level correlations, which results in serious accumulated decoding errors and prevents users from efficient parallel decoding with fewer timesteps. To this end, in this paper, we propose Energy-based Diffusion Language Model (EDLM), an unnormalized energy-based model (EBM) that learns to jointly denoise the full sequence at each diffusion step. Our key innovation is to learn an EBM for each denoising distribution $p(\mathbf{x}_{t-1}|\mathbf{x}_t)$, where the energy directly operates on the sequence level and captures the correlation between tokens. The fundamental challenge of our framework is to develop efficient training and sampling methods for the unnormalized model. To overcome these challenges, we design the EBM in a novel residual form

$$p_{\text{EDLM}}(\mathbf{x}_{t-1}|\mathbf{x}_t) = p_{\text{diffusion}}(\mathbf{x}_{t-1}|\mathbf{x}_t) \exp(-E(\mathbf{x}_{t-1}, \mathbf{x}_t)), \tag{1}$$

applied over pretrained diffusion models $p_{\text{diffusion}}$. Such formulation enjoys several distinct advantages. First, we analytically show that the EBM parameters can be easily obtained by leveraging pretrained autoregressive models or finetuning from bidirectional transformers via noise contrastive estimation, bypassing expensive maximum likelihood training. Furthermore, the framework corrects the decoding error and enables fast generation by conducting efficient important sampling in parallel over samples from the diffusion proposal distribution. Importantly, we highlight that, when leveraging pretraiend AR as the energy function, our approach can be interpreted as parallel sampling from pretrained language models using diffusion models as the proposal distribution.

EDLM can be viewed as a new family of discrete generative models that marries energy-based and diffusion-based models. The novel combination addresses the training and sampling distribution mismatch problem, enjoys better generation quality with less accumulated error, and improves sampling efficiency by reducing the number of sampling steps. We further provide a formal estimator to calculate the perplexity of EDLM, allowing our model to be compared against other models in a standard way. We conduct comprehensive experiments on two common language modeling benchmarks to evaluate the performance of our proposed method. Results show that on the perplexity metric, EDLM can consistently achieve state-of-the-art performance among diffusion-based counterparts, and approaches or matches AR models. On the generation quality metric, compared over the most competitive diffusion baseline, EDLM shows up to $49\%$ generative perplexity gain with the same number of sampling timesteps, and can achieve up to $1.3\times$ sampling speedup when keeping the same sampling performance.

## 2 RELATED WORK

Diffusion models are powerful models with surprising results in generating high-quality images (Sohl-Dickstein et al., 2015; Song & Ermon, 2019; Ho et al., 2020; Dhariwal & Nichol, 2021). These models were originally designed for continuous data, with both forward and backward reverse processes parameterized as Gaussian Markov chains. In recent years, several methods have been developed to extend the diffusion framework to generate discrete data (Austin et al., 2021; Hoogeboom et al., 2022), with Campbell et al. (2022); Zhao et al. (2024) further extending the framework to model discrete data in continuous time. Most recently, several concurrent works (Lou et al., 2024; Sahoo et al., 2024; Shi et al., 2024) archived impressive progress in large-scale language modeling by scaling the model and simplifying the training and sampling processes. Additionally, Campbell et al. (2024); Gat et al. (2024) developed flow matching methods for discrete data, which rely on a formulation similar to that of diffusion models. Despite the considerable progress in the area, we observe that all these approaches are subject to the training and sampling distribution mismatch issue, where the learned joint denoising distribution is simplified as independent distributions for each token. This problem results in a significant accumulated decoding error during the parallel sampling process and prevents users from conducting fast sampling with a small number of denoising timesteps. Deng et al. (2020) studies similar accumulated error problem (*a.k.a* exposure bias) in the autoregressive model setting and proposes energy-based models for modeling global context, but the methodology focuses on improving autoregressive models and therefore fundamentally different from ours. Other work (Gu et al., 2018; Ghazvininejad et al., 2019; Gu & Kong, 2021; Savinov et al., 2022; Zheng et al., 2023) study similar non-autoregressive text generation but combined with reranking and/or remasking methods. However, these methods typically result in biased sampling, which can generate high-quality samples but fail to model the distribution and suffer from

low sampling diversity. Therefore, these methods mainly focus on different tasks such as machine translation. Notably, Lezama et al. (2023) notices a similar issue (focusing on diffusion models in the image generation domain) and proposes to learn a corrector network to correct the independent decoding error. However, this work concentrates on the image generation domain, and the correction process needs to be run sequentially, which takes additional inference time. In contrast, we propose an energy-based denoising process that directly captures the correlation at each denoising step and enables efficient parallel decoding via an importance sampling scheme.

## 3 DISCRETE DIFFUSION MODELS

**General Discrete Diffusion**. Assume our data $\mathbf{x}$ lives in a finite discrete space of size $m - 1$. In this paper, we augment the categorical space with an additional mask state with index $m$. In a general discrete diffusion framework (Austin et al., 2021), the diffusion process is defined as a Markov chain $q(\mathbf{x}_t|\mathbf{x}_{t-1}) = \text{Cat}(\mathbf{x}_t; Q_t\mathbf{x}_{t-1})$, which repeatedly multiplies $\mathbf{x}$ with matrices $Q_t$ over $T$ discrete time steps. Given these transitions, the marginal distributions at each timestep can be written in closed-form as $q(\mathbf{x}_t|\mathbf{x}) = \text{Cat}(\mathbf{x}_t; \bar{Q}_t\mathbf{x}) = \text{Cat}(\mathbf{x}_t; Q_t \cdots Q_1\mathbf{x})$. Such forward process can also be viewed as an interpolation between a clean data sample $\mathbf{x}$ and a reference distribution $\text{Cat}(\cdot; \pi)$ induced by $\bar{Q}_T$:

$$q(\mathbf{x}_t|\mathbf{x}) = \text{Cat}(\mathbf{x}_t; \alpha_t\mathbf{x}_0 + (1 - \alpha_t)\pi), \tag{2}$$

where $\alpha_t \in [0, 1]$ is a strictly decreasing function *w.r.t* $t$, with $\alpha_0 \approx 1$ and $\alpha_1 \approx 0$.

In the continuous time limit, for two arbitrary times $0 \leq s \leq t \leq 1$, the transition distributions can be written as $q(\mathbf{x}_t|\mathbf{x}_s) = \text{Cat}(\mathbf{x}_t; \alpha_{t|s}\mathbf{x}_s + (1 - \alpha_{t|s})\pi)$, where $\alpha_{t|s} = \alpha_t/\alpha_s$ (Zhao et al., 2024; Shi et al., 2024). This implies that during each diffusion step $s \to t$ the token will jump to a sample from the prior distribution $\pi$ with a probability of $(1 - \alpha_{t|s})$. The forward process allows us to compute many distributions in closed form. One particular distribution of interest is the reversal of the forward process given $\mathbf{x}_0$, that is, the posterior distribution given by

$$q(\mathbf{x}_s|\mathbf{x}_t, \mathbf{x}_0) = \text{Cat}\left(\mathbf{x}_s; \frac{[\alpha_{t|s}\mathbf{x}_t + (1 - \alpha_{t|s})\mathbf{1}\pi^\top\mathbf{x}_t] \odot [\alpha_s\mathbf{x}_0 + (1 - \alpha_s)\pi]}{\alpha_t\mathbf{x}_t^\top\mathbf{x}_0 + (1 - \alpha_t)\mathbf{x}_t^\top\pi}\right). \tag{3}$$

**Masked Diffusion**. In this paper, we focus on masked (*i.e.*, absorbing state) diffusion models, where the target distribution is set as $\pi = \mathbf{m}$. At each diffusion step $t$ each token transitions to the 'masked' state $\mathbf{m}$ with some probability. Under such masking framework, the forward marginals (Eq. (2)) are given by $q(\mathbf{x}_t|\mathbf{x}_0) = \alpha_t\mathbf{x}_0 + (1 - \alpha_t)\mathbf{m}$, and the posterior (Eq. (3)) can be simplified as

$$q(\mathbf{x}_s|\mathbf{x}_t, \mathbf{x}_0) = \begin{cases} \text{Cat}(\mathbf{x}_s; \mathbf{x}_t) & \mathbf{x}_t \neq \mathbf{m}, \\ \text{Cat}\left(\mathbf{x}_s; \frac{(1-\alpha_s)\mathbf{m}+(\alpha_s-\alpha_t)\mathbf{x}_0}{1-\alpha_t}\right) & \mathbf{x}_t = \mathbf{m}. \end{cases} \tag{4}$$

Diffusion models aim to learn a backward model $p_\theta(x_s|x_t)$ to approximate the reversal of the forward process. Leveraging Eq. (4) we can parameterize the model $p_\theta(\mathbf{x}_s|\mathbf{x}_t)$ as

$$p_\theta(\mathbf{x}_s|\mathbf{x}_t) = q(\mathbf{x}_s|\mathbf{x}_t, \mathbf{x}_0 = \boldsymbol{\mu}_\theta(\mathbf{x}_t, t)) = \begin{cases} \text{Cat}(\mathbf{x}_s; \mathbf{x}_t), & \mathbf{x}_t \neq \mathbf{m}, \\ \text{Cat}\left(\mathbf{x}_s; \frac{(1-\alpha_s)\mathbf{m}+(\alpha_s-\alpha_t)\boldsymbol{\mu}_\theta(\mathbf{x}_t, t)}{1-\alpha_t}\right), & \mathbf{x}_t = \mathbf{m}, \end{cases} \tag{5}$$

where $\boldsymbol{\mu}_\theta = p_\theta(\mathbf{x}_0|\mathbf{x}_t)$ is known as $\mathbf{x}_0$ predictor, since it predicts the mean of the distribution over $\mathbf{x}_0$ given $\mathbf{x}_t$. However, as explained next, the practical implementation of this $\mathbf{x}_0$ predictor results in a mismatch between training and sampling distributions.

**Problem Statement**. Existing discrete DMs learn a denoising distribution $\boldsymbol{\mu}_\theta = p_\theta(\mathbf{x}_0|\mathbf{x}_t)$ to match the true reversal $q(\mathbf{x}_0|\mathbf{x}_t)$. Let $\mathbf{x}$ denote the tokens of a full sequence in this section. In practice, the model is parameterized as a factorized denoising model:

$$p_\theta(\mathbf{x}_0|\mathbf{x}_t) = \Pi_i p_\theta(\mathbf{x}_0^i|\mathbf{x}_t) = \Pi_i \boldsymbol{\mu}_\theta^i(\mathbf{x}_t, t), \text{ where } \boldsymbol{\mu}_\theta(\mathbf{x}_t, t) = \begin{cases} \text{softmax}(f_\theta(\mathbf{x}_t, t)) & \mathbf{x}_t = \mathbf{m}, \\ \mathbf{x}_t & \mathbf{x}_t \neq \mathbf{m}, \end{cases} \tag{6}$$

and the predictor $\boldsymbol{\mu}_\theta$ factorizes each token in $\mathbf{x}_0$ independently. This factorization enables us to conduct an efficient denoising step $p_\theta(\mathbf{x}_s|\mathbf{x}_t)$ by first sampling all $\mathbf{x}_{0|t}$ tokens from $p_\theta(\mathbf{x}_0|\mathbf{x}_t)$ in

parallel and then masking certain tokens according to the forward $q(\mathbf{x}_s|\mathbf{x}_t, \mathbf{x}_{0|t})$. However, this parameterization ignores dependencies between tokens in the sequence, a fundamental limitation which implies that $p_\theta(\mathbf{x}_0|\mathbf{x}_t)$ can never match the exact backward $q(\mathbf{x}_0|\mathbf{x}_t)$. As a result, this parallel sampling introduces accumulated errors, as the factorized denoising step $p_\theta$ does not match the original generative model $p_\theta$ for the joint distribution of the elements of $\mathbf{x}_0$.

## 4 METHOD

In this section, we formally introduce Energy-based Diffusion Language Model (EDLM). Our work aims to design a new family of energy-based discrete generative models that address the fundamental mismatch problem between $p_\theta(\mathbf{x}_0|\mathbf{x}_t)$ and $q(\mathbf{x}_0|\mathbf{x}_t)$ in existing models. We first describe the general EBM formulation in Section 4.1, and then elaborate on how to obtain the energy function by either leveraging pretrained AR models or finetuning via noise contrastive estimation in Section 4.2. Then we discuss how to estimate the likelihood (perplexity) of EDLM in Section 4.3, and finally introduce the efficient parallel sampling algorithm in Section 4.4

### 4.1 RESIDUAL ENERGY-BASED MODELS

Discrete DMs can be viewed as learning a conditional $\mathbf{x}_0$ predictor for each denoising step $t$. For example, original discrete DMs learn $\boldsymbol{\mu}(\mathbf{x}_t, t)$ to directly predict independent distributions for each token in $\mathbf{x}_0$. In our framework, given diffused data $\mathbf{x}_t$ at timestep $t$, we introduce the generative denoising kernel as an unnormalized density:

$$p_{\theta,\phi}(\mathbf{x}_0|\mathbf{x}_t) = \boldsymbol{\mu}_\theta(\mathbf{x}_0|\mathbf{x}_t) \frac{\exp(-\boldsymbol{E}_\phi(\mathbf{x}_0, \mathbf{x}_t, t))}{Z_\phi(\mathbf{x}_t)} \tag{7}$$

where $\boldsymbol{\mu}_\theta$ is the pretrained diffusion model, $\boldsymbol{E}_\phi$ is the energy introduced to capture the correlation in the $\mathbf{x}_0$ sequence, and $Z_\phi(\mathbf{x}_t)$ is a normalizing factor known as the partition function. In the following text, we use $p_{\theta,\phi}$ to denote the joint model, $\boldsymbol{E}_\phi$ the (residual) energy function, and keep the pretrained $\boldsymbol{\mu}_\theta$ fixed. Computing the partition function is intractable since it requires summing over the whole $\mathbf{x}$ space, which is exponential in the sequence length. In our language modeling experiments, the vocabulary size is around 50k and the generation length is $1024$ tokens, resulting in a space of size around $50,000^{1024}$. We aim to design solutions to efficiently train the parameters of the energy function so that the joint model distribution gets close to the true reversal $q(\mathbf{x}_0|\mathbf{x}_t)$, while avoiding computing the partition function.

### 4.2 IMPLEMENTATION OF ENERGY FUNCTION

Training energy-based models with the intractable partition function is a long-standing challenge in machine learning (Hinton, 2002; Carreira-Perpinan & Hinton, 2005; LeCun et al., 2006). Typical maximum likelihood estimation training requires approximation of the participation function using Markov chain Monte Carlo (MCMC) sampling, which is computationally infeasible for high-dimensional data. We aim to find efficient ways to train the energy function's parameters for the $\mathbf{x}_0$ predictor $p_{\theta,\phi}(\mathbf{x}_0|\mathbf{x}_t)$. In the following, we describe two methods to do this. One involves taking pretrained autoregressive language models as energy functions without any training, and efficiently running sampling by taking all tokens as inputs in parallel. The second solution involves fine-tuning the pretrained diffusion model via noise contrastive estimation, where the model is parameterized with bidirectional transformers and potentially captures richer correlations.

#### 4.2.1 LEVERAGE PRE-TRAINED AUTOREGRESSIVE MODELS

Let $p_{\text{AR}}(\mathbf{x}_0)$ be an autoregressive model trained over clean samples $\mathbf{x}_0$, *i.e.*, $p_{\text{AR}}(\mathbf{x}_0) = \Pi_i p_{\text{AR}}(\mathbf{x}_0^i|\mathbf{x}_0^{<i})$. Since AR models are only defined over clean samples $\mathbf{x}_0$, it is not directly clear how to leverage them in denoising tasks, since the models are never trained on diffused data $\mathbf{x}_t$. Our key insight is that, in absorbing discrete diffusion, the diffused $\mathbf{x}_t$ is the same as $\mathbf{x}_0$ with certain tokens masked according to the forward process. More specifically, letting $\bar{\mathbf{x}}_0 = \mathbf{x}_0[\mathbf{x}_t \neq \mathbf{m}]$ and $\mathbf{x}_0/\bar{\mathbf{x}}_0$ denote the set of $\mathbf{x}_0$ tokens corresponding to unmasked and masked components in $\mathbf{x}_t$, we

observe that we can induce the denoising transitions using

$$p_{\text{AR}}(\mathbf{x}_0 \,|\, \mathbf{x}_t) = p_{\text{AR}}(\mathbf{x}_0 \,|\, \bar{\mathbf{x}}_0) = \frac{p_{\text{AR}}(\mathbf{x}_0/\bar{\mathbf{x}}_0, \bar{\mathbf{x}}_0)}{p_{\text{AR}}(\bar{\mathbf{x}}_0)} = \frac{p_{\text{AR}}(\mathbf{x}_0)}{p_{\text{AR}}(\bar{\mathbf{x}}_0)}, \tag{8}$$

where the normalizing constant is given by $p_{\text{AR}}(\bar{\mathbf{x}}_0) = \sum_{\mathbf{x}_0/\bar{\mathbf{x}}_0} p_{\text{AR}}(\mathbf{x}_0/\bar{\mathbf{x}}_0, \bar{\mathbf{x}}_0)$. This partition function also involves a sum over the $\mathbf{x}_0$ space and is intractable to compute. However, noting that in the reversal of masked diffusion (Eq. (5)) unmasked tokens $\bar{\mathbf{x}}_0$ are fixed, we have that $p_{\text{AR}}(\mathbf{x}_0 \,|\, \mathbf{x}_t) \propto p_{\text{AR}}(\mathbf{x}_0)$,[1] which can be computed efficienctly.

Unlike the original diffusion models (Eq. (6)) that decode each token independently, AR models factorize the sequence distribution and take into account dependencies between tokens. Therefore, it is reasonable to assume AR models can provide a better approximation of the diffusion posterior than the factorized approximation used by diffusoin models, *i.e.*, $p_{\text{AR}}(\mathbf{x}_0 \,|\, \mathbf{x}_t) \approx q(\mathbf{x}_0 \,|\, \mathbf{x}_t)$. The key challenge is that *sampling* the autoregressive model $p_{\text{AR}}(\mathbf{x}_0 \,|\, \mathbf{x}_t)$ at each denoising step is computationally expensive. However, it is efficient to *evaluate* $p_{\text{AR}}(\mathbf{x}_0)$, which is proportional to the posterior $p_{\text{AR}}(\mathbf{x}_0 \,|\, \mathbf{x}_t)$, and thus can be used in our residual energy-based formulation.

Recall that our residual energy model (Eq. (7)) uses $p_\theta(\mathbf{x}_0 \,|\, \mathbf{x}_t)\frac{\exp(-\boldsymbol{E}_\phi(\mathbf{x}_0, \mathbf{x}_t))}{Z_\phi(\mathbf{x}_t)}$ to approximate $q(\mathbf{x}_0 \,|\, \mathbf{x}_t)$. By taking $p_{\text{AR}}(\mathbf{x}_0 \,|\, \mathbf{x}_t)$ as the proxy of $q(\mathbf{x}_0 \,|\, \mathbf{x}_t)$, we approximate the optimal residual energy as

$$\begin{aligned}
\boldsymbol{E}_\phi(\mathbf{x}_0, \mathbf{x}_t) &= -\log q(\mathbf{x}_0|\mathbf{x}_t) + \log p_\theta(\mathbf{x}_0 \,|\, \mathbf{x}_t) - \log Z \\
&\approx -\log p_{\text{AR}}(\mathbf{x}_0|\mathbf{x}_t) + \log p_\theta(\mathbf{x}_0 \,|\, \mathbf{x}_t) - \log Z \qquad\quad q(\mathbf{x}_0 \,|\, \mathbf{x}_t) \approx p_{\text{AR}}(\mathbf{x}_0|\mathbf{x}_t) \\
&= -\log p_{\text{AR}}(\mathbf{x}_0) + \log p_\theta(\mathbf{x}_0 \,|\, \mathbf{x}_t) + \underbrace{\log p_{\text{AR}}(\bar{\mathbf{x}}_0) - \log Z}_{\text{New Partition Function}} \qquad \text{See Eq. (8).}
\end{aligned}$$
$$\tag{9}$$

For common EBM model inference, such as MCMC, we only require energies up to a constant. Therefore, as shown in the equation above, with a pretrained AR model we can simply define the energy function as $-\log p_{\text{AR}}(\mathbf{x}_0) + \log p_\theta(\mathbf{x}_0 \,|\, \mathbf{x}_t)$. The fact that AR language models can evaluate likelihoods in parallel enables the development of efficient sampling algorithms. We leverage this fact to develop a sampling method based on self-normalized importance sampling using $p_\theta(\mathbf{x}_0 \,|\, \mathbf{x}_t)$ as the proposal distribution (see Section 4.4). Importantly, we highlight that this energy formulation translates to sampling from the AR language model as the target distribution using the denoising distribution as the proposal distribution, providing a novel way to conduct parallel sampling from pretrained AR language models via importance sampling.

**Carry-Over (CO) Parameterization**. Note that in diffusion reversal (Eq. (4)), unmasked tokens are always directly carried over from $\mathbf{x}_t$ to $\mathbf{x}_0$, *i.e.*, $q(\mathbf{x}_0|\mathbf{x}_t) = \text{Cat}(\cdot; \mathbf{x}_t)$ for $\mathbf{x}_t \neq \mathbf{m}$. Recent progress in discrete diffusion models show that directly parameterizing this property into denoising model may improve the generation performance. In practice this is implemented by substituting the output of the $\boldsymbol{\mu}_\theta$ network to simply copy the unmasked tokens of $\mathbf{x}_t$. Similarly, in our AR EBM model, we can also carry over the unmasked tokens in each denoising step, by directly setting $p_{\text{AR}}(\mathbf{x}_0^i|\mathbf{x}_0^{<i}) = \mathbf{x}_t^i$ for $\mathbf{x}_t \neq \mathbf{m}$. This "carry-over autoregressive" (coAR) EBM implementation allows simply computing $p_{\text{coAR}}(\mathbf{x}_0 \,|\, \mathbf{x}_t) = \frac{p_{\text{coAR}}(\mathbf{x}_0)}{p_{\text{coAR}}(\bar{\mathbf{x}}_0)}$ by just $p_{\text{coAR}}(\mathbf{x}_0)$, as we always carry over those tokens from $\mathbf{x}_t$ and therefore $p_{\text{coAR}}(\bar{\mathbf{x}}_0) = 1$. Such practice enables us to evaluate the exact denoising likelihood without estimating the partition function. We will introduce how to calculate the data likelihood based on denoising distribution in Section 4.3.

### 4.2.2 TRAINING WITH NOISE CONTRASTIVE ESTIMATION

We can also train the parameters of the residual energy function using Noise Contrastive Estimation (NCE) (Gutmann & Hyvärinen, 2010), specifically its conditional version (Ma & Collins, 2018). First, NCE training relies on contrastive samples from the data distribution and a noise distribution, where the noise distribution needs to be close to the data distribution. Second, it involves computing the likelihoods of these samples by model distribution and noise distribution. In our approach, given a pair of clean data $\mathbf{x}_0$ and diffused data $\mathbf{x}_t$, we set true posterior $q(\hat{\mathbf{x}}_0|\mathbf{x}_t, \mathbf{x}_0)$[2] as positive

---

[1]This property does not require the autoregressive structure of $p_{\text{AR}}$, but only relies on the fact that during the diffusion reversal process unmasked tokens do not change.

[2]In this subsection, we use notation $\hat{\mathbf{x}}_0$ instead $\mathbf{x}_0$ to distinguish the generated data and clean data $\mathbf{x}_0$.

distribution and the denoising distribution $p_\theta(\hat{\mathbf{x}}_0|\mathbf{x}_t)$ as the negative distribution. Thanks to the residual energy formulation from Eq. (7), the log-odds reduce to $\log p_{\theta,\phi} - \log p_\theta = -\boldsymbol{E}_\phi$, and the objective is simplified into the binary classification objective

$$\mathcal{L}_{\text{NCE}}(\phi;\theta) = -\mathbb{E}_{\mathbf{x}_0 \sim p_{\text{data}}, \mathbf{x}_t \sim q(\mathbf{x}_t|\mathbf{x}_0)} \Bigg[$$
$$\mathbb{E}_{\mathbf{x}_+ \sim q(\hat{\mathbf{x}}_0|\mathbf{x}_t,\mathbf{x}_0)} \log \frac{1}{1 + \exp(\boldsymbol{E}_\phi(\mathbf{x}_+, \mathbf{x}_t, t))} + \mathbb{E}_{\mathbf{x}_- \sim p_\theta(\hat{\mathbf{x}}_0|\mathbf{x}_t)} \log \frac{1}{1 + \exp(-\boldsymbol{E}_\phi(\mathbf{x}_-, \mathbf{x}_t, t))} \Bigg], \tag{10}$$

where $\mathbf{x}_+$ is positive data from the true posterior and $\mathbf{x}_-$ is negative data from diffusion model, given the diffused sequence $\mathbf{x}_t$. Here, the true posterior $q(\hat{\mathbf{x}}_0|\mathbf{x}_t, \mathbf{x}_0) := \mathbf{x}_0$ is defined as simply recovering the true data. Training the energy function can be viewed as training a conditional classifier to discriminate the real text and text generated by the denoiser used by the diffusion model. Intuitively, the training objective attempts to capture the correlation in $\mathbf{x}_0$ generations, assigning negative energy to real data and positive energy to data produced by the denoiser network. As a result, the joint model $p_{\theta,\phi}$ acts as a corrected denoising distribution able to take into account correlations between tokens. A detailed pseudo code for the NCE training process is provided in Algorithm 2.

## 4.3 Evaluation with Rao-Blackwellized Likelihood Bounds

A common protocol for evaluating discrete generative models on language modeling is perplexity (PPL), which relies on computing log-likelihoods for a hold-out test dataset. In this section, we explain how to estimate the log-likelihood with EDLM. First, with the energy-based $\mathbf{x}_0$ predictor $p_{\theta,\phi}(\mathbf{x}_0|\mathbf{x}_t)$, the step denoising model $p_{\theta,\phi}(\mathbf{x}_s|\mathbf{x}_t)$ is given by

$$p_{\theta,\phi}(\mathbf{x}_s|\mathbf{x}_t) = \mathbb{E}_{p_{\theta,\phi}(\mathbf{x}_0|\mathbf{x}_t)} q(\mathbf{x}_s|\mathbf{x}_t, \mathbf{x}_0)$$
$$= \frac{1 - \alpha_s}{1 - \alpha_t} \text{Cat}(\mathbf{x}_s; \mathbf{m}) + \frac{\alpha_s - \alpha_t}{1 - \alpha_t} p_{\theta,\phi}(\mathbf{x}_0[\mathbf{x}_s \neq \mathbf{m}] = \mathbf{x}_s[\mathbf{x}_s \neq \mathbf{m}]|\mathbf{x}_t) \tag{11}$$

For the simplified reverse process in Eq. (5), the log-likelihood can be simply computed as a form of Rao-Blackwellization. The Rao-Blackwellized likelihood bound can be estimated by the discrete-time diffusion loss of finite $T$: $\mathcal{L}_{\text{diffusion}} = \sum_{i=1}^{T} \mathbb{E}_q[D_{\text{KL}}(q(\mathbf{x}_{s_i}|\mathbf{x}_{t_i}, \mathbf{x})\|p_{\theta,\phi}(\mathbf{x}_{s_i}|\mathbf{x}_{t_i}))]$. Under EDLM, the loss can be written as:

$$\mathcal{L}_{\text{diffusion}} = \sum_{i=1}^{T} \mathbb{E}_q \left[ \frac{\alpha_{t_i} - \alpha_{s_i}}{1 - \alpha_{t_i}} \left( \log p_\theta(\mathbf{x}_0|\mathbf{x}_{t_i}) - \boldsymbol{E}_\phi(\mathbf{x}_0, \mathbf{x}_{t_i}, t_i) - \log Z_\phi(\mathbf{x}_{t_i}) \right) \right], \tag{12}$$

where $p_\theta(\mathbf{x}_0|\mathbf{x}_{t_i}) = \langle \boldsymbol{\mu}_\theta(\mathbf{x}_{t_i}), \mathbf{x} \rangle$ is a simple cross-entropy between $\boldsymbol{\mu}_\theta$ predicted logits and the clean data. We can extend the objective Eq. (12) to the continuous limit by taking $T \to \infty$, which induces the following negative evidence lower bound (NELBO) $\mathcal{L}_\infty$:

$$\mathcal{L}_\infty = \mathbb{E}_q \int_{t=0}^{t=1} \frac{\alpha_t'}{1 - \alpha_t} \left( \log p_\theta(\mathbf{x}_0|\mathbf{x}_t) - \boldsymbol{E}_\phi(\mathbf{x}_0, \mathbf{x}_t, t) - \log Z_\phi(\mathbf{x}_t) \right) \mathrm{d}t. \tag{13}$$

where $\alpha_t'$ denotes the derivative of $\alpha_t$ w.r.t $t$. Such ELBO is tighter than the discrete-time version, and is invariant to the noise schedule. However, this bound requires the partition function $\log Z_\phi(\mathbf{x}_t) = \log \sum_{\mathbf{x}_0} p_\theta(\mathbf{x}_0|\mathbf{x}_t) \exp(-\boldsymbol{E}_\phi(\mathbf{x}_0, \mathbf{x}_t)) = \log \mathbb{E}_{\mathbf{x}_0 \sim p_\theta} \exp(-\boldsymbol{E}_\phi(\mathbf{x}_0, \mathbf{x}_t))$ and thus is intractable to compute. We use two estimators based on the work (Nowozin, 2018; Deng et al., 2020) to estimate the partition function $\log Z_\phi$.

**Theorem 1.** *Given diffused data $\mathbf{x}_t$ at timestep $t$, let $\log \mathcal{Z}_n$ denote the empirical estimation of $\log Z_\phi(\mathbf{x}_t) = \log \mathbb{E}_{\mathbf{x}_0 \sim p_\theta} \exp(-\boldsymbol{E}_\phi(\mathbf{x}_0, \mathbf{x}_t))$ with $n$ samples $\mathbf{x}_0^{(i)} \sim p_\theta(i = 1, \cdots, n|\mathbf{x}_t)$: $\log \mathcal{Z}_n = \log \frac{1}{n} \sum_{i=1}^{n} \exp(-\boldsymbol{E}(\mathbf{x}_0^{(i)}, \mathbf{x}_t))$. Then $\forall \epsilon > 0, \exists N > 0$ such that $\forall n > N$ we have*

$$\log Z_\phi - \epsilon < \mathbb{E}[\log \mathcal{Z}_n] < \log Z_\phi < \mathbb{E}[(2n-1)\log \mathcal{Z}_n - 2(n-1)\log \mathcal{Z}_{n-1}] < \log Z_\phi + \epsilon. \tag{14}$$

This can be used to estimate lower and upper bounds of the partition function. We note, however, that the bounds only hold asymptotically when $n$ is sufficiently large. In practice, we follow Nowozin (2018) to use the leave-one-out strategy to estimate $\mathcal{Z}_{n-1}$, which is proven to yield an estimator with lower variance. In addition, we want to recall that in our autoregressive energy function instance (section 4.2.1), we introduce the carry-over parameterization, where the joint energy model is self-normalized, which enables us to exactly compute the ELBO without estimating $Z_\phi$.

### 4.4 Efficient Generation via Importance Sampling

Sampling from EDLM with each denoising step operated by the joint model is non-trivial. Naive approaches relying on Gibbs sampling (Gelfand, 2000; Hinton, 2002) were originally applied to binary inputs and are not scalable to large dictionaries with energy functions parameterized as large transformer models. In this paper, we resort to self-normalizing importance sampling (Owen, 2013; Hammersley, 2013). With our joint model as the product of the diffusion model $p_\theta$ and residual energy function $\boldsymbol{E}_\phi$, and given intermediate diffusion data $\mathbf{x}_t$ at timestep $t$, we can conduct efficient parallel sampling by: 1) sampling multiple $\mathbf{x}_0$ predictions $\{\mathbf{x}_0^i\}_{i=1}^k$ from the diffusion denoiser $p_\theta(\mathbf{x}_0|\mathbf{x}_t)$; 2) feeding samples into energy function in parallel to compute energies; and 3) resampling a single $\mathbf{x}_0$ from the pool $\{\mathbf{x}_0^i\}_{i=1}^k$ according to the energy values. The sampled $\mathbf{x}_0$ is then fed into the posterior formulation $q(\mathbf{x}_s|\mathbf{x}_t, \mathbf{x}_0)$ to perform one reverse step (*i.e.* one-step denoising).

**Importance Sampling Window**. While importance sampling introduces additional computation, it yields a significant reduction in the parallel decoding error. Therefore, the method allows us to conduct diffusion sampling with fewer denoising steps, reducing the overall sampling wall-clock time. To further accelerate the sampling speed, we introduce the concept of importance sampling window length $w \in [0,1]$, which sets the timestep for stopping importance sampling, *i.e.*, we only conduct importance sampling in the time window $t \in [1-w, 1]$. Interestingly, in our experiment, we notice that the energy-based importance sampling during the early stage of denoising sampling contributes more to the generation quality improvement. We conclude that this phenomenon is due to the fact that during the early stage of sampling the diffusion model is prone to make more errors in independent $\mathbf{x}_0$ prediction, since there is little information on the full sequence. This encourages us to explore sampling with a short time window w for higher efficiency, which we discuss in detail in section 5.3.

---

**Algorithm 1** Denoising via Importance Sampling

1: **Input:** discrete diffusion model $\theta$, energy-based model $\phi$, sequence of timesteps $\tau_1 > \tau_2 > \cdots > \tau_{N-1}$, number of sampling size $k$, importance sampling window w
2: $\mathbf{x}_{\tau_1} \leftarrow \mathbf{m}$
3: **for** $n = 1$ **to** $N-1$ **do**
4: $\quad p_\theta(\mathbf{x}_0|\mathbf{x}_{\tau_n}) \leftarrow \boldsymbol{\mu}_\theta(\hat{\mathbf{x}}_{\tau_n})$
5: $\quad$ **if** $\tau_n \geq 1 - \mathrm{w}$ **then**
6: $\qquad \{\mathbf{x}_0^1, \cdots, \mathbf{x}_0^k\} \sim p_\theta(\mathbf{x}_0|\mathbf{x}_{\tau_n})$
7: $\qquad$ Compute energies $\mathbf{e}^i = \boldsymbol{E}_\phi(\mathbf{x}_0^i, \mathbf{x}_{\tau_n})$
$\qquad\qquad\qquad$ for $\mathbf{x}_0^i \sim \{\mathbf{x}_0^1, \cdots, \mathbf{x}_0^k\}$
8: $\qquad$ Sample $\mathbf{x}_0 \sim \{\mathbf{x}_0^1, \cdots, \mathbf{x}_0^k\}$
$\qquad\qquad$ with probability $\frac{\exp(-\mathbf{e}^i)}{\sum_{j=1}^k \exp(-\mathbf{e}^j)}$
9: $\quad$ **else**
10: $\qquad$ Sample $\mathbf{x}_0 \sim p_\theta(\mathbf{x}_0|\mathbf{x}_{\tau_n})$
11: $\quad$ **end if**
12: $\quad \mathbf{x}_{\tau_{n+1}} \sim q(\mathbf{x}_{\tau_{n+1}}|\mathbf{x}_{\tau_{n+1}}, \mathbf{x}_0)$
13: **end for**
14: **Output:** $\mathbf{x} = \mathbf{x}_{\tau_N}$

---

Detailed pseudo code for the sampling procedure is provided in Algorithm 1. In practice, we observe that a relatively small importance sampling size can already yield significant performance gain, though theoretically we would only recover exact samples from the joint model distribution asymptotically when the number of importance samples goes to infinity.

## 5 Experiments

This section presents the results achieved by our method on different language modeling tasks. We compare our EDLM against existing diffusion models, reporting the widely adopted metrics perplexity and generative perplexity. We provide the experimental setup in section 5.1, describe our results in section 5.2, and provide additional ablation studies in section 5.3.

### 5.1 Experimental Setup

**Datasets**. We use two text datasets: 1) Text8 (Mahoney, 2006), a relatively small-scale, character-level text modeling benchmark extracted from English Wikipedia, and 2) OpenWebText, an open-source replica of the unreleased WebText (Gokaslan & Cohen, 2019) dataset used to train GPT-2.

**Baselines**. We compare EDLM against state-of-the-art discrete diffusion models and transformer AR modelS (Vaswani et al., 2017). Discrete diffusion baselines include Discrete Diffusion Model (D3PM) (Austin et al., 2021), Score Entropy Discrete Diffusion (SEDD) (Lou et al., 2024), Masked

Diffusion Language Model (MDLM) (Sahoo et al., 2024) and MD4 (Shi et al., 2024). We notice other recent works such as discrete flow matching (Campbell et al., 2024; Gat et al., 2024), but they are either not open-sourced or focus on non-language settings. Therefore, we take their concurrent work with similar formulation MDLM (Sahoo et al., 2024) and MD4 (Shi et al., 2024) as the representative for comparison. On the small-scale Text8 benchmark, we additionally evaluate other discrete generative models including Plaid (Gulrajani & Hashimoto, 2024), Bayesian Flow Network (Graves et al., 2023), Any-order Autoregressive Models ARDM (Hoogeboom et al., 2022) and MAC (Shih et al., 2022), and flow-based methods IAF/SCF (Ziegler & Rush, 2019), AR Argmax Flow (Hoogeboom et al., 2021), Discrete Flow (Tran et al., 2019), and Multinomial Diffusion (Hoogeboom et al., 2021). Note that, these methods are not scaled up yet to the large dictionary and long sequence length, and thus we only involve them in the small-scale benchmark.

**Metrics**. We follow the convention (Shi et al., 2024) to use the common protocols Bits Per Character (BPC), Perplexity (PPL), and Generative Perplexity (Gen PPL) to evaluate generative sequence models. For a sequence of length $L$, BPC metric is given by $-\frac{1}{L}\sum_{i=1}^{L}\log_2 p(\mathbf{x}_i)$ and PPL is given by $\exp\left(-\frac{1}{L}\sum_{i=1}^{L}\log p(\mathbf{x}_i)\right)$, which can be viewed as the average number of tokens the model is uncertain of. BPC and PPL are calculated based on model likelihoods on true data from the test set. Gen PPL instead consists of likelihoods calculated by another large oracle model on data generated by the evaluated models. Intuitively, PPL and BPC similarly evaluate the likelihood modeling capacity, while Gen PPL evaluates generation quality and consistency. To compute Gen PPL we generate 2048 samples of 1024.

**Implementation Details**. For all models including our methods and baselines, we follow the common practice of using standard 12-layer transformers similar to GPT2-small scale (Radford et al., 2019; Shi et al., 2024). Our proposed EDLM combines two models, the diffusion model $p_\theta$ and the energy function $\mathbf{E}_\phi$. For all experiments, we use pretrained MDLM (Sahoo et al., 2024) as the diffusion model $p_\theta$. For AR-based EBM (see section 4.2.1, named **EDLM-AR** or **EDLM-coAR** when using carry-over), we directly leverage the pretrained AR model as the energy function. For NCE fine-tuned EBM (see section 4.2.2, named **EDLM-NCE**), we finetune the pretrained MDLM, with the energy function computed by projecting the mean-pooled last token embeddings down to a single scalar value. Note that, MDLM relies on transformers with bidirectional attention while AR only imposes casual attention. Therefore, we expect that EDLM-NCE can capture more sequence information than EDLM-AR.

Table 1: Bits Per Character (BPC) on Text8 test set.

| Method | BPC ($\downarrow$) |
|---|---|
| *Autoregressive* | |
| Transformer AR | **1.23** |
| IAF/SCF | 1.88 |
| AR Argmax Flow | 1.39 |
| AR Discrete Flow | **1.23** |
| *Any-order Autoregressive* | |
| ARDM | $\leq$ 1.43 |
| MAC | $\leq$ 1.40 |
| *Continuous Diffusion* | |
| Plaid | $\leq$ 1.48 |
| BFN | $\leq$ 1.41 |
| *Discrete Diffusion* | |
| Mult. Diffusion | $\leq$ 1.72 |
| D3PM Uniform | $\leq$ 1.61 |
| D3PM Absorb | $\leq$ 1.45 |
| SEDD Absorb | $\leq$ 1.41 |
| MDLM | $\leq$ 1.40 |
| MD4 | $\leq$ 1.37 |
| EDLM (Ours) | $\leq$ **1.24** |

## 5.2 RESULTS

**Text8.** Following the previous common practice (Austin et al., 2021; Lou et al., 2024), we evaluated all models on short text chunks of length 256. We follow the same dataset splits and transformers model size to parameterize the denoising models. Results are summarized in Table 1, where we report the standard bits-per-character metric for the Text8 test set. In this small-scale experiment, we did not notice significant differences between EDLM-AR and EDLM-NCE, and therefore only report one result as representative. As shown in the table, EDLM outperforms all previous diffusion models, whether in discrete or continuous diffusion formulation. We also outperform the any-order autoregressive models, which also generate sequences with flexible decoding order and therefore have a strong connection to discrete diffusion models. Importantly, for the first time, our diffusion model even approaches the performance of transformer AR and AR Discrete Flow, both relying on autoregressive modeling. For this experiment, due to the limited dictionary and length size, we did not observe meaningful signals in generation quality metrics (Gen PPL). We leave detailed generation quality and diversity evaluation to the large-scale OpenWebText experiments next.

**OpenWebText**. We report the perplexity results of all models trained on the large-scale OpenWebText dataset in Table 2. We evaluate the model capacity for both the OpenWebText test set and

Table 2: Test perplexities (↓). Left part: results evaluated on the OpenWebText test set; Right part: zero-shot results on unseen datasets. All perplexities for diffusion models are upper bounds.

| | OpenWebText | PTB | Wikitext | LM1B | Lambda | AG News | Pubmed | Arxiv |
|---|---|---|---|---|---|---|---|---|
| AR | 17.56 | 82.05 | 25.75 | 51.25 | 51.28 | 52.09 | 49.01 | 41.73 |
| SEDD | 24.56 | 100.09 | 34.28 | 68.20 | 49.86 | 62.09 | 44.53 | 38.48 |
| MLDM | 23.83 | 95.26 | 32.83 | 67.01 | 47.52 | 61.15 | 41.89 | 37.37 |
| EDLM-NCE (Ours) | 21.52 | 93.21 | 30.77 | 63.19 | **46.92** | 60.02 | **41.80** | **36.63** |
| EDLM-AR (Ours) | 20.49 | **89.67** | 29.24 | 60.80 | 49.70 | **57.27** | 45.90 | 38.38 |
| EDLM-coAR (Ours) | **17.58** | 89.73 | **28.31** | **60.23** | 50.04 | 57.94 | 46.31 | 39.02 |

[*] Results of baseline methods on unseen datasets are borrowed from (Sahoo et al., 2024).

Table 3: Generative perplexities on unconditional text generation. LLAMA2, LLAMA3, and GPT2 (↓) correspond to Generative perplexities evaluated by different oracle models.

| Method | Timesteps | LLAMA2↓ | LLAMA3↓ | GPT2↓ | Entropy |
|---|---|---|---|---|---|
| Data | - | 7.0 | 9.4 | 14.7 | 7.7 |
| Autoregressive | 1024 | 22.9 | 40.3 | 35.7 | 8.1 |
| SUNDAE | 200 | 29.5 | 45.1 | 34.7 | 5.2 |
| Ssd-LM | >10000 | 73.3 | 203.1 | 99.2 | 4.8 |
| D3PM Absorb | 1024 | 692.3 | 754.9 | 842.3 | 7.6 |
| SEDD | 256 / 512 / 1024 / 2048 | 36.1 / 32.5 / 27.3 / 23.1 | 65.0 / 54.3 / 43.7 / 36.2 | 64.8 / 52.2 / 41.5 / 33.7 | 7.8 / 7.7 / 7.6 / 7.5 |
| MDLM | 256 / 512 / 1024 / 2048 | 37.2 / 30.6 / 27.6 / 23.9 | 66.8 / 52.6 / 44.6 / 37.6 | 66.8 / 52.4 / 42.6 / 34.9 | 7.9 / 7.8 / 7.6 / 7.5 |
| EDLM-AR (Ours) | 256 / 512 / 1024 / 2048 | **34.7** / 26.8 / 19.6 / **14.6** | **62.2** / 44.4 / **28.8** / 20.8 | 62.1 / **42.0** / 25.5 / 17.9 | 7.9 / 7.6 / 7.2 / 6.9 |
| EDLM-NCE (Ours) | 256 / 512 / 1024 / 2048 | 35.7 / **26.3** / **19.0** / **14.6** | 62.9 / **44.1** / **28.8** / 20.7 | **61.7** / 42.5 / **25.5** / **17.7** | 7.9 / 7.6 / 7.3 / 6.9 |

[*] Results of SUNDAE and Ssd-LM are borrowed from (Gat et al., 2024).

seven out-of-domain unseen datasets, used to validate the models' ability in zero-shot generalization. These unseen datasets include Our Penn Tree Bank (PTB) (Marcus et al., 1993)), Wikitext (Merity et al., 2016), LM1B, Lambada (Paperno et al., 2016), AG News (Zhang et al., 2015), and Scientific Papers (Pubmed and Arxiv) (Cohan et al., 2018). As shown in the results, we again observe that EDLM outperforms existing diffusion methods by a significant margin, and also approaches the AR baseline. For EDLM-NCE and EDLM-AR, since the energy function is unnormalized, we use the upper bound estimator in Theorem 1 to estimate the upper bound for negative log-likelihood, which induces the upper bounds of perplexities shown in the table. For EDLM-coAR, as discussed in Section 4.2.1, we overcome the estimation of the partition function when combined with carry-over and thus compute the exact ELBO for likelihoods.

We also compare our method against prior non-autoregressive generative models for generation quality. All models are trained on OpenWebText, and results are presented in Table 3. We additionally incorporate Step-unrolled Denoising Autoencoder (SUNDAE) (Savinov et al., 2022) and Ssd-LM (Han et al., 2022) into the comparison. These two methods are also highly related diffusion methods for text generation which cannot be assessed by log-likelihood. In this experiment, we set the importance sampling window w = 1, *i.e.* use importance sampling at every denoising step, to evaluate the full generation ability of EDLM. Additionally, EDLM-AR shows similar results either with or without the carry-over trick, therefore we only report one method for simplicity. As shown in Table 3, our method again consistently outperforms all baselines on generative perplexity for all numbers of denoising timesteps, while keeping reasonable diversity based on the entropy metric. We highlight that our method does not hurt the entropy (diversity), since it achieves similar entropy as MDLM when keeping similar Gen PPL. For example, by comparing EDLM with 512 timesteps and MDLM with 1024 steps, we can see they show roughly the same Gen PPL and entropy score, but EDLM requires significantly fewer sampling steps to reach that level of performance.

## 5.3 ANALYSIS AND ABLATION STUDY

**Improved Sampling Efficiency**. In previous experiments, we show that with the same number of timesteps sampling, EDLM can achieve significantly better generation results with less accumulated error. In this part, we highlight that EDLM can also enable better sampling efficiency when keeping the same generative quality, by using reduced denoising steps. As introduced in section 4.4, our key observation is that the energy-based importance sampling correction during the early stage of denoising sampling contributes more to the generative perplexity improvement. Therefore, to accelerate the sampling speed, in this experiment, we set the importance sampling size as $k = 2$ window

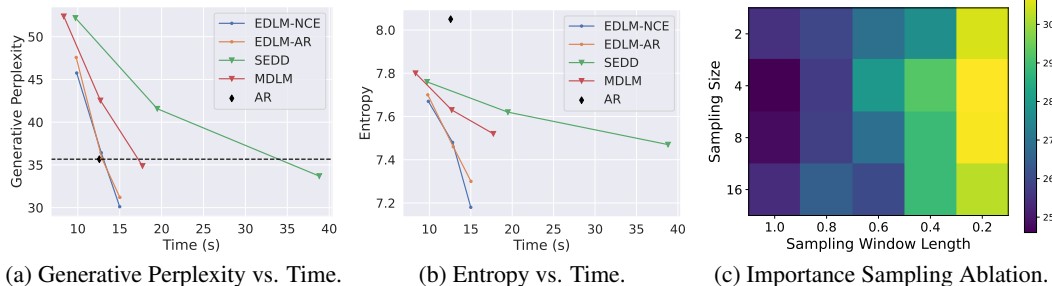

(a) Generative Perplexity vs. Time.    (b) Entropy vs. Time.    (c) Importance Sampling Ablation.

Figure 1: Analysis and ablation study for EDLM. Figures 1a and 1b: we run AR baseline and diffusion-based models with $[512, 768, 1024]$ denoising steps, and plot the curve of corresponding metric vs. wall-clock time. For generative perplexity, the metric is evaluated by GPT-2, and a curve on the bottom-left indicates a better sampling quality vs time trade-off. Figure 1c: ablation study of EDLM under different importance sampling size and window length.

as w $= 0.2$, which empirically yields the best generation quality with the same wall-clock sampling time budget. w $= 0.2$ means we use importance sampling for $t \in [0.8, 1]$, and just use the original diffusion model $\boldsymbol{\mu}_\theta$ for denoising for $t \in [0, 0.8]$. We provide additional results with other window size in Appendix C.2. We test AR baseline and diffusion-based models with $[512, 768, 1024]$ denoising steps, and plot the curve of corresponding metric vs. wall-clock time for generating a single 1024-length sentence in Figures 1a and 1b. As shown in the figure, EDLM achieves a better sampling quality vs time trade-off compared against MDLM. Specifically, when achieving a similar generative perplexity as the AR baseline (35.7), we can see from the figure that EDLM only takes ∼13 seconds while the MDLM baseline takes ∼17 seconds, indicating a ∼1.3× acceleration without any performance drop. This speedup further highlights the advantage of our energy-based denoising formulation and importance sampling scheme, where we can conduct sampling with reduced denoising steps and correct the decoding errors by parallel importance sampling.

**Effect of Importance Sampling**. We further study the effect of importance sampling hyperparameters in this section. We fix the number of denoising timesteps as 1024, and investigate the generation performance with varying importance sampling size and window length. We refer readers to Section 4.4 and Algorithm 1 for details of these two hyperparameters. Again, we found that EDLM-NCE and EDLM-AR exhibit similar trends in the study, and therefore show one study result here. The ablation study results are summarized in Figure 1c, from which we highlight two key observations: 1) Row-wise comparison indicates that the sampling quality is not sensitive to the importance sampling size. However, we emphasize that the conclusion is based on sizes smaller than 16, as a larger size will lead to out-of-memory issues on the GPU. 2) Column-wise study show that a longer importance sampling window can consistently improve the sampling quality. However, we note here that when setting the window as 0, the sampling decays to the original MDLM sampling, and the 1024 steps GPT-2 Gen PPL is 42.6 (see Table 3). Therefore, a 0.2 window already significantly reduces the Gen PPL to ∼ 30 while a longer window only marginally further improves the results. This phenomenon suggests that major decoding error happens during early-stage denoising sampling, and a short window can greatly overcome the errors while maintaining high sampling efficiency, as shown in our efficiency study above.

## 6    CONCLUSION

In this paper, we introduced Energy-based Diffusion Language Model (EDLM), which integrates energy-based models with discrete diffusion models to address the limitations of parallel text generation. By leveraging a residual energy-based approach, EDLM effectively reduces the mismatch between training and sampling distributions in discrete diffusion models, resulting in improved generation quality and efficiency. Through experiments on both small and large language modeling benchmarks, EDLM demonstrates state-of-the-art performance among diffusion models and approaches the quality of autoregressive models, while offering significant sampling speedup. These results highlight the potential of energy-based approaches in advancing discrete generative modeling, setting the stage for further exploration of parallel generation techniques.

## REPRODUCIBLITY STATEMENT

We provide detailed pseudo codes of relevant algorithms in Algorithm 1 and Algorithm 2. We provide the formal proof for theorem 1 in Appendix A. We describe the details for used datasets, data processing, model architecture, training and evaluation implementations in Section 5.1 and appendix C.1. We emphasize transparency and reproducibility for machine learning research.

## ACKNOWLEDGMENT

MX thanks the generous support of Sequoia Capital Stanford Graduate Fellowship. SE gratefully acknowledge the support of ARO (W911NF-21-1-0125), ONR (N00014-23-1-2159), and Chan Zuckerberg Biohub. JL gratefully acknowledge the support of NSF under Nos. OAC-1835598 (CINES), CCF-1918940 (Expeditions), DMS-2327709 (IHBEM), IIS-2403318 (III); Stanford Data Applications Initiative, Wu Tsai Neurosciences Institute, Stanford Institute for Human-Centered AI, Chan Zuckerberg Initiative, Amazon, Genentech, GSK, Hitachi, SAP, and UCB.

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

# A PROOF

To make the paper self-contained, we incorporate the relevant theoretical proof for Theorem 1 from (Nowozin, 2018; Deng et al., 2020). Our key difference is correcting a minor misargument from estimation on $Z_\phi$ to estimation on $\log Z_\phi$.

**Theorem 1.** *Given diffused data $\mathbf{x}_t$ at timestep $t$, let $\log \mathcal{Z}_n$ denote the empirical estimation of $\log Z_\phi(\mathbf{x}_t) = \log \mathbb{E}_{\mathbf{x}_0 \sim p_\theta} \exp(-\boldsymbol{E}_\phi(\mathbf{x}_0, \mathbf{x}_t))$ with $n$ samples $\mathbf{x}_0^{(i)} \sim p_\theta (i = 1, \cdots, n | \mathbf{x}_t)$: $\log \mathcal{Z}_n = \log \frac{1}{n} \sum_{i=1}^n \exp(-\boldsymbol{E}(\mathbf{x}_0^{(i)}, \mathbf{x}_t))$. Then $\forall \epsilon > 0, \exists N > 0$ such that $\forall n > N$ we have*

$$\log Z_\phi - \epsilon < \mathbb{E}[\log \mathcal{Z}_n] < \log Z_\phi < \mathbb{E}[(2n-1) \log \mathcal{Z}_n - 2(n-1)\mathcal{Z}_{n-1}] < \log Z_\phi + \epsilon, \quad (15)$$

*Proof.* Following Eq. 35 in Nowozin (2018), we have $\mathbb{E}[\log \mathcal{Z}_n]$ as

$$\mathbb{E}[\log \mathcal{Z}_n] = \log Z_\theta - \frac{\gamma_2}{2\gamma^2}\frac{1}{n} + \frac{1}{3\gamma^3}\frac{\gamma_3}{n^2} - \frac{1}{4\gamma^4}(\frac{3}{n^2}\gamma_2^2 + \frac{1}{n^3}(\gamma_4 - 3\gamma_2^2)) + \frac{1}{5\gamma^5}(\frac{10}{n^3}\gamma_3\gamma_2 + \frac{1}{n^4}(\gamma_5 - 10\gamma_3\gamma_2)) + o(n^{-3})$$
(16)

where $\gamma = \mathbb{E}[\log \mathcal{Z}_n], \gamma_k = \mathbb{E}[(\log \mathcal{Z}_n - \gamma)^k]$.

By omitting the higher-order expansion, we equivalently have $\mathbb{E}[\log \mathcal{Z}_n] = \log Z_\theta - \frac{\gamma_2}{2\gamma^2}\frac{1}{n} + o(n^{-1})$, which indicates that $\lim_{n\to\infty} \mathbb{E}[\log \mathcal{Z}_n] = \log Z_\theta$. Then, we have

- $\forall \epsilon > 0, \exists N_1 > 0$ such that for $n > N_1$, $\mathbb{E}[\log \mathcal{Z}_n] > \log Z_\theta - \epsilon$.

- Since $\lim_{n\to\infty} n(\log Z_\theta - \mathbb{E}[\log \mathcal{Z}_n]) = \lim_{n\to\infty} \frac{\gamma_2}{2\gamma^2} + o(1) = \frac{\gamma_2}{2\gamma^2} > 0$, so $\exists N_2 > 0$ such that for $n > N_2$, $\log Z_\theta > \mathbb{E}[\log \mathcal{Z}_n]$.

In summary, we have proved that $\log Z_\theta - \epsilon < \mathbb{E}[\log \mathcal{Z}_n] < \log Z_\theta$.

Again, by omitting higher-order expansions using Eq. (16), we can have another equivalent form. $\mathbb{E}[\log \mathcal{Z}_n] = \log Z_\theta - \frac{\gamma_2}{2\gamma^2}\frac{1}{n} + \frac{c}{n^2} + o(n^{-2})$ where $c$ is a constant, and therefore $\mathbb{E}[(2n-1)\log \mathcal{Z}_n - 2(n-1)\log \mathcal{Z}_{n-1}] = (2n-1)\mathbb{E}[\log \mathcal{Z}_n] - 2(n-1)\mathbb{E}[\log \mathcal{Z}_{n-1}] = \log Z_\theta + \frac{\gamma_2}{2\gamma^2}\frac{1}{n} + o(n^{-1})$. This indicates $\lim_{n\to\infty} \mathbb{E}[(2n-1)\log \mathcal{Z}_n - 2(n-1)\log \mathcal{Z}_{n-1}] = \log Z_\theta$. Therefore, we have

- $\forall \epsilon > 0, \exists N_3 > 0$ such that $\forall n > N_3$ $\mathbb{E}[(2n-1)\log \mathcal{Z}_n - 2(n-1)\log \mathcal{Z}_{n-1}] < \log Z_\theta + \epsilon$.

- Since $\lim_{n\to\infty} n(\mathbb{E}[(2n-1)\log \mathcal{Z}_n - 2(n-1)\log \mathcal{Z}_{n-1}] - \log Z_\theta) = \lim_{n\to\infty} \frac{\gamma_2}{2\gamma^2} + o(1) > 0$, so $\exists N_4 > 0$ such that when $n > N_4$, $\mathbb{E}[(2n-1)\log \mathcal{Z}_n - 2(n-1)\log \mathcal{Z}_{n-1}] > \log Z_\theta$.

Given all the results above, we have that $\forall \epsilon > 0$, for $\forall n > N$ that $N = \max\{N_1, N_2, N_3, N_4\}$

$$\log Z_\theta - \epsilon < \mathbb{E}[\log \mathcal{Z}_n] < \log Z_\theta < \mathbb{E}[(2n-1)\log \mathcal{Z}_n - 2(n-1)\log \mathcal{Z}_{n-1}] \log Z_\theta + \epsilon$$

□

# B ALGORITHM

We provide the detailed pseudo code for noise contrastive training of the EBM in Algorithm 2.

---

**Algorithm 2** Noise Contrastive Estimation Training

---

1: **Input:** dataset $\mathcal{D}$, discrete diffusion model $\theta$, energy-based model $\phi$, learning rate $\eta$
2: **repeat**
3:     Sample $\mathbf{x}_0 \sim \mathcal{D}$ and $t \sim U[0,1]$
4:     $\mathbf{x}_t \sim q(\mathbf{x}_t | \mathbf{x}_0)$
5:     Sample positive data $\mathbf{x}_+ \sim q(\cdot | \mathbf{x}_t, \mathbf{x}_0)$
6:     Sample negative data $\mathbf{x}_- \sim p_\theta(\cdot | \mathbf{x}_t)$
7:     $\mathcal{L}(\phi; \theta) \leftarrow \mathcal{L}_{\text{NCE}}(\mathbf{x}_+, \mathbf{x}_-)$ in Eq. (10)
8:     $\phi \leftarrow \phi - \eta \nabla_\phi \mathcal{L}(\phi; \theta)$
9: **until** convergence

---

## C  ADDITIONAL EXPERIMENT DETAILS AND RESULTS

### C.1  ADDITIONAL EXPERIMENT DETAILS

We provide additional experiment setup details in this section.

**Text8**. We follow all the common practices in Austin et al. (2021); Campbell et al. (2024) to conduct Text8 experiments, which has a dictionary size of 28 with 26 lowercase letters, a white-space token, and a mask token. We follow the standard dataset split and train MDLM using a standard 12-layer transformer architecture. the transformer also has the same number of heads (12) and hidden dimension (784) as in Austin et al. (2021). The model is trained text chunks of length 256 for 1 million steps with batch size 512. In the NCE setting, we then finetune from this MDLM with a pooling layer and an additional scalar energy prediction head. We observe the finetuning procedure to converge fast in just $10,000$ steps. For both MDLM training and EDLM-NCE finetuning, we follow previous work to use the cosine learning rate schedule with a linear warm-up of 2000 steps. We set the channel-wise dropout rate as $0.05$ and conducted optimization with AdamW and a learning rate of $0.0003$. We similarly adopt a weight decay factor of $0.03$. The NCE finetuning process can be done on 4 GPUs for less than 4 hours.

**OpenWebText**. We follow the standard data split in (Sahoo et al., 2024) to leave a validation split with the last 100k docs as the validation set. We tokenize OpenWebText with the GPT2 tokenizer, with a vocabulary size of around 50K. All models are trained with sequences wrapped to a length of $1,024$ and additionally set eos as the first and last token of every batch. All the architectural choices are kept the same with the Text8 experiment, where we use transformers with 12 layers, a hidden dimension of 768, 12 attention heads, and a timestep embedding of 128 when applicable. Word embeddings are not tied between the input and output. Other training details are also kept the same, *i.e.*, we use the AdamW optimizer with a batch size of 512, learning rate $0.0003$ with a linear warm-up of 2500 steps. We train all models for 1M steps with the dropout rate reduced to $0.1$. Again, the NCE finetuning of the energy function from the pretrained diffusion model is efficient and can converge in $400,000$ steps.

### C.2  ADDITIONAL EXPERIMENT RESULTS

We provide additional experimental results in this section.

**Behavior of Energy Function**. We first provide a closer look at the behavior of energy function across different denoising steps. Specifically, for each timestep $t$, we draw diffused data from $q(\mathbf{x}_t|\mathbf{x}_0)$, and then sample positive data $\mathbf{x}_+ \sim q(\cdot|\mathbf{x}_t, \mathbf{x}_0)$ and negative data $\mathbf{x}_- \sim p_\theta(\cdot|\mathbf{x}_t)$, similar to Algorithm 2. Then we study the energy value for $E_\phi(\mathbf{x}_+, \mathbf{x}_t)$ and $E_\phi(\mathbf{x}_-, \mathbf{x}_t)$.

We summarize all plots for the energy *w.r.t* timestep in Figure 2. We sample 16 negative samples for the visualization. Specifically, in Figure 2, in the first row, we plot the energy of positive samples and the average energy of negative samples, to see whether the energy function can differentiate true and generate data; in the second row, we plot the maximum and minimum energy value of the 16 negative samples, to see whether energy function can differentiate better and worse samples in generated data; and in the third row, we plot the effective sampling size (ESS) for energies of the 16 negative samples. Formally, let $\mathbf{e}^i$ denote the energy for $i$-th negative sample, we first normalize the energies to $\hat{\mathbf{e}}^i = \frac{\mathbf{e}^i}{\sum_{i=1}^{16} \mathbf{e}^i}$, and then ESS is given by

$$\text{ESS} = \frac{(\sum_{i=1}^{16} \hat{\mathbf{e}}^i)^2}{\sum_{i=1}^{16} (\hat{\mathbf{e}}^i)^2}, \tag{17}$$

which intuitively helps quantify the degeneracy of the sample weights in importance sampling. A low ESS indicates that a few samples dominate the weights, which can lead to poor estimation quality. Different columns of Figure 2 correspond to results for different EDLM parameterization. From the first two rows, we can see all EDLM implementations can assign meaningful energies to true data, good generated sample, and bad generated sample. For ESS, it indicates that the NCE energy function is generally better than AR ones for importance sampling. However, we note that we did not observe that EDLM-NCE is clearly better than EDLM-AR, and we conclude that this is mainly because our ESS analysis is conducted with 16 samples instead of an extremely large

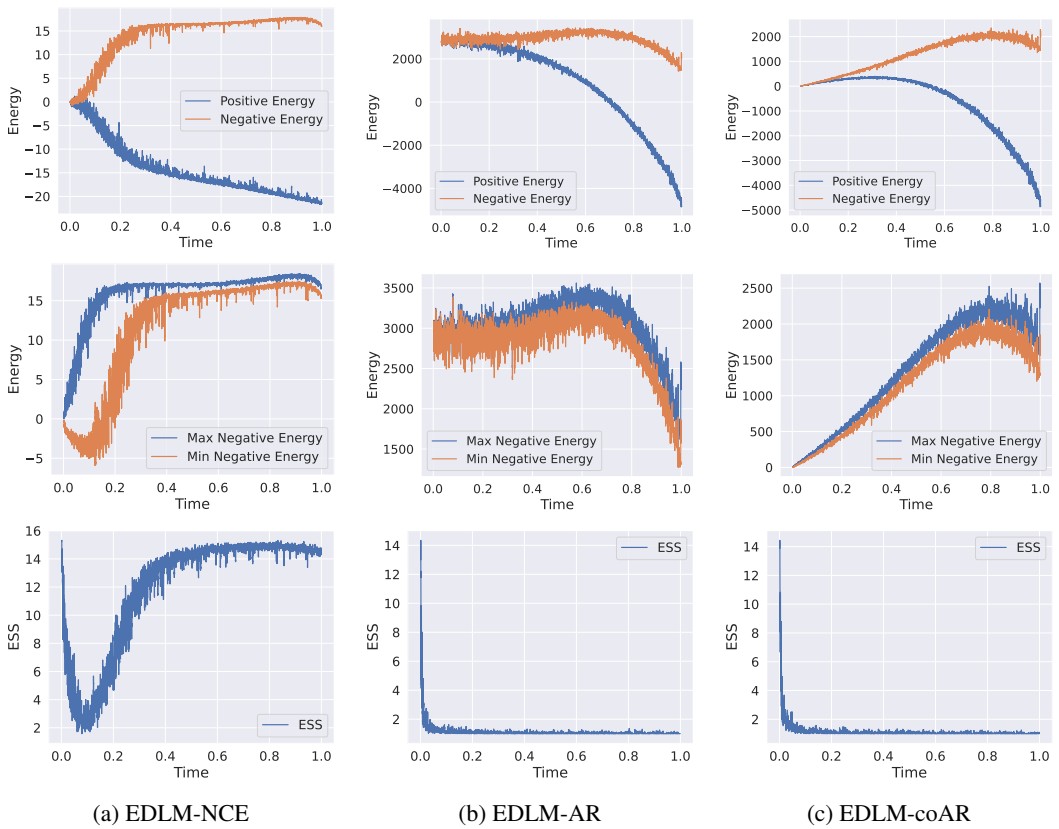

Figure 2: Behavior of the energy function under different parameterization. The first row plots the energy of positive samples and the average energy of negative samples; the second row plots the maximum and minimum energy values of the 16 negative samples; the third row plots the effective sampling size (ESS) for energies of the 16 negative samples. Different columns correspond to results for different EDLM parameterization.

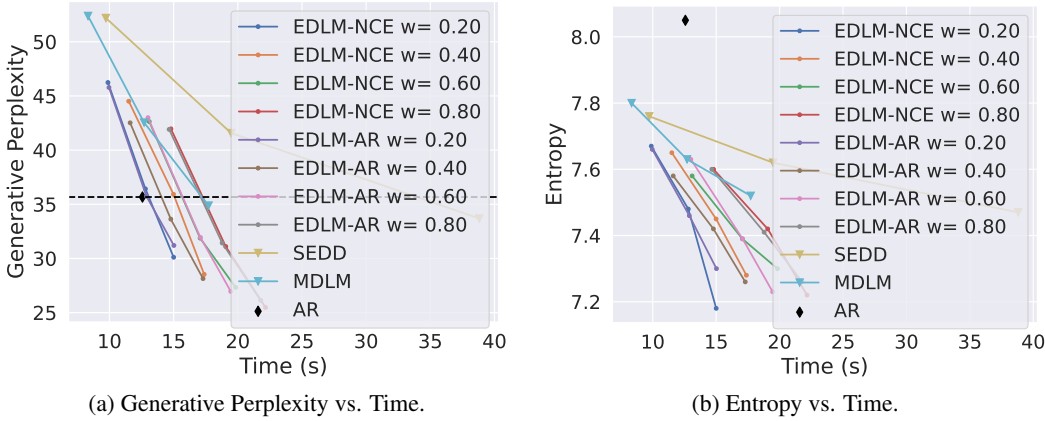

Figure 3: Additional EDLM sampling results with varying importance sampling window $\mathbf{w}$. Similar to section 5.3, we run each setting with $[512, 768, 1024]$ denoising steps, and plot the curve of the corresponding metric vs. wall-clock time. For generative perplexity, the metric is evaluated by GPT-2, and a curve on the bottom-left indicates a better sampling quality vs time trade-off.

number. We recall here that GPU memory bound the sampling batch size to 16, and therefore large ESS analysis in practice is not useful.

**Additional results with different sampling window length w.** We provide additional sampling results with different importance sampling window sizes w. All results are provided in Figure 3. As shown in the figure, especially the reference line of AR in Figure 3a, we can see that generally $w \leq 0.6$ can all lead to better efficiency, among which $w = 0.2$ achieve the best speedup.

**Additional visualization of Gen PPL and entropy results.** We provide a more detailed trade-off between entropy and Gen PPL compared to the baselines and the proposed model. Specifically, we rearrange Figures 1a and 1b to a new figure with Entropy and Gen PPL axis, and provide the figure in Figure 3b. As shown in the figure, EDLM shows a trade-off similar to MDLM ($< 0.1$ difference of entropy in the overlapping part). However, EDLM aims to reduce the sampling errors and therefore has a similar performance as MDLM but with much fewer sampling timesteps.

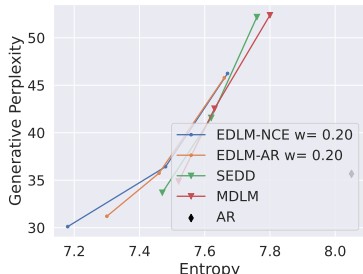

Figure 4: Gen. PPL vs. Entropy.

# D  GENERATED SAMPLES FROM EDLM

## D.1  GENERATED SAMPLES FROM EDLM-NCE WITH 1024 TIMESTEPS.

```
...a lot. I remember when you was a kid, you felt like you weren't
a human. And I said, \You didn't wanna get into that." \Yeah,
that's just physically [grown-up], I kind of weighed down with
that.\n\nWell, yeah, trying to help him, but it was something
to give. Not the little stuff. It was tough to get from an
older person. It just kinda won. That's the way with things like
that.\n\nBut you had to learn something else, more of appreciation
of it, watching guys play [inside out] as you grew. Everything
they showed up you knew was always in the way they wanted, as just
as your parents wanted you.\n\nAnd [my mom] when I was yelling
at her: \But you get the opportunity to work something for each
other, and you can cook us meals as well." \Yep.\n\nWe had to
find some way out. It got borderline ridiculous, but we found a
way.\n\nHe and I speak and we talk a lot. That's really the thing
about us. Just the two dogs at the right time.\n\nNot funny man.
Just like we are, baby. We're not stupid."\n\n\What happened like
when you're getting bumps?"\n\n\No, I can\'t see it from that side
of the eye. I wish I could. I can\'t see it on my face. Yeah, I
don\'t really. I picked it up and it's not the right way. But the
way I got it."\n\nMarcus Cousins\n\nI think I'm not confused. He
was so close to me. But I'm never confused either.\n\nAre you
playing the same type of game now?"\n\n\Always the same way. You
play it at the same time. You don't have the same way before you
play the game. And I got used to it. He had a good season. He was
a defensive superstar.\n\n\Sometimes we would go through the type
of moves when I's saying things, be like, 'Oh, he'm cheating on
me,' or something like that because we just know like that. Lord,
you know, stuff. That's part of this life. Like my dad and I had
a little bit, and that got you sometimes it didn't work to the
liking. [You] would physically get him.\n\nWe did, but he always
demanded that I catch up to him. And when I tell him, \Catch me
if you say something stupid, you're not stupid." If he does say
stupid something, you know, I'm going to respond to it."\n\nIt's
like \What\'s going to do catch \'n\' it? If he's gonna catch it,
that\'s my one."\n\nAnything like that?\n\n\It\'s funny because
a lot of people have a playing style that plays the player they
are. He's a caricature of what it is, but he's a kind guy and
it affects the game."\n\n\That's a good one.\n\n\His skills
that he play, can he tell you?"\n\n\Yes. He's an unbelievable
```

dude.\n\nYeah, he's not the guy that everybody goes to play with, just out on the team. He plays stuff because he takes the game seriously. But you see a player that goes and develop that's easy to feed off of, and that's good. They win games, too, by being a teammate, in that environment.\n\nBut that's all.\n\nHonestly, it hasn't been really easy for me. I played 35 percent of the time and these guys ask a lot about me [laughs]. I told you guys those guys. I know that when I was young I would not have made it. I was just born that way when I was there. So don\'t call me the only man that I really liked or the backbone of the team, ever. That guy is the person in the locker.\n\nWhen you're all of the guys in your team, you don't have to make credit for what you do.\n\nBut I definitely miss Tom or Cole. Everybody does that to me. I love Chad Mac. It's the Strict style. He\'s always teaching me on being an open guy.\n\nThat kind of stuff, what I like the most. If I lose a lot of games,...

## D.2 Generated Samples from EDLM-AR with 1024 timesteps.

...have to make you better. But all you have to do, you can go talk about that with him. If you want it to accomplish that from his position as the coach, then you definitely feel that as a manager will have a tremendous benefit as a player, and that's very important, that is, why hockey. That's what you want to achieve as an organization. So, when you get to manage, where the truth is that you're better as a player and the more points you score and the more passes you have to make and make plays more often, you feel that it's enhanced your effectiveness as a player.\n\nRNS: And you can do that?\n\nJPP: Totally. I just think you can. So, you've gotta know, if you're going to play that role of a manager, you'll be the one that knows where to go. And that's how great it is, every year.\n\nRNS: If it's difficult as a manager in your last one, did you find yourself right about that decision? Did you feel better?\n\nQ \Well, I sat there the other day thinking, 'OK, I have to do this job to get a career in that position.' So when I do that: one-thousand up, no down. I could not go on. I didn't do what I need to say about that and I can't help but be proud of the success that I have created." And I think, \You just gotta realize that you're not supposed to truly have your own voice." \I can't do that' 'I let the people out of my decisions,' then I think you are one of that.\n\nBut you can go through that. Just to live in this is to live in having to really utilize this and offer this to you. So, if you're to move out of this job, where you want to move out of this job when you are a player, where maybe when you gotta be manager, when you gotta do business, or when you are a hockey player, you decide which is different. So we're the same positions but we talk about how to do it.\n\nAnd just, more and more we've said: that's how you're gonna get there. If you want that manager role, you should do this. Because you've got to be passionate about your passion for your side, frontend your team up and do everything that takes to build this team as an organization. And they're players. So you don't get to go around and have to do that. If you don't do that, you'll have a lot of control of your players on the team. You also've got to help them and show them what they're doing, to be involved. To take care of and make sure that their players succeed. So, you have to be productive within the team. That's two things.\n\nRNS: So what level of coaching is interesting and what do you hope to get out of all of those roles

and positions?\n\nJPP: Yeah, you never know. And you know how they're actually putting these things. So when you start a coach, you're in to young players, you're in with multiple players every day, you coach to the players who are currently in a role in an organization, and obviously that helps build them. So basically, your really good coaching means you'll wear your jersey, be the best player you can be from your experiences. I always tell people after you're done doing the things, you trust me that it's the most important to take place next.\n\nSo, when I started coaching, I do not think that was a great leap because that was my first time in the NHL. But I would like to tell the coaches, \You're gonna stick with your team, you'll lead the team. Let's be with the best," and I assume, the next coach, you'll be the one and you'll be the one and that season you'll be the one. And that's what you gotta think about is that if you're more involved and in unique ways, if you're learning from your experience and will be successful as importantly, just, you know, put their teams on the table. That's part of what we are trying to do. And I'd say I'd like to put people on that front that are strong and fit for the job. So that it...

### D.3 Generated Samples from EDLM-coAR with 1024 timesteps.

...to explain that to someone who has not shared information with Google itself and not the powers that be. And Kim didn't.\n\n\I never wanted to do this because before it, I didn't know anything. Finally, you feel as if you do."\n\nI laughed.\n\n\It didn't feel right."\n\n\But yeah, we stayed together for a while," Kim said, \and when we had time for us to, we were talking."\n\nI was almost in silence. \I'm not able to act out that way," she said, \so I just kept having my time. One day when I was traveling and rehearsing with my own band, we had a really funny meeting. It's funny, it was just plain funny."\n\n\That day, we were talking to other girls friends and friends and I had my first year-long meeting with social media guys. As a women's boss."\n\n\I feel like different company here," she said. \I mean, it's basically an out-of-work company."\n\n\I had to pick up my car at the same time when we first met."\n\n\Oh yeah. It was weird, because I did a lot with her, and she didn't know my love for her while I was in her way. Odd. She was sitting right right next to me at my cousin's house, hanging out with friends."\n\nWhen she was speaking to North, she felt in touch with friends and family in Marlboro where I lived, where she raised me and how we still hadn't familiar with each other since as far back as I went and saw how she had got confidence over him when I was 18. I didn't, too, but she said he took everything else into account. Even with Wests death this morning, I can still have a private conversation with Kim with everyone. Some conversations on the phone and for the other girls.\n\nKim leaned in very talkily and he said to her, \We don't like you much," but he wasn't trying to make California's jealous. \You're the one if you're engaged to somebody and if you see someone else, it's probably never good for you, so don't go out to anything that's not good for you." And that's all too true.\n\nI first asked Kim West about his faith in him as a woman's friend, what he thought of moving to Marlboro and how that goes him down.\n\n\I'm a guy that is always doing something great," West had said. \And then I'll come up short, do well myself, and I go, 'I really want someone on this,'" she said.\n\n\That's sad," Jörg smiled, \this kind of guy who's not that smart and not do well. I

mean, you looked the fuck you down and you figured out that's more important to me than you thought you were, basically. You wanted to do something."\n\n\It's sad," he said. \That was all I had my years trying to figure out about you so that I could link you and spend time with somebody who's gonna roll."\n\n\I could just read his shit and die," she said, gesturing to West's letters. \And that's how I was left behind. It's always struck me that I didn't know how I wanted to spend with this group or another."\n\n\Had you dated her in the past?" she asked, asking if he had more of a problem with her? \No, not really," he said. \I'm a nice guy and I would talk back to anyone and that is why I actually talk to people."\n\n\That would be a really creepy place to be in," he told her, adding, \it was like...the first time I've lost a friend."\n\nThere was blood in the room. \Lemmer, he was a funny dude at the time."\n\nHe smiled. \But you know that. This way a happy friend would have been if I had him back then. I didn't care for him. ..."

