# OpenReview forum: "Energy-Based Diffusion Language Models for Text Generation"
_ICLR.cc/2025/Conference — ICLR 2025 Poster_

### Official Review · Reviewer_wiWZ · 2024-10-21

**Soundness:** 3
**Presentation:** 3
**Contribution:** 2
**Rating:** 5
**Confidence:** 4

**Summary:**

This paper points out that the existing discrete diffusion models aim to predict all missing tokens in parallel at each intermediate diffusion step, but the denoising joint distribution is simply parameterized as a product of token-wise independent distributions.
The authors propose Energy-based Diffusion Language Model (EDLM) by using an energy-based model to act as a reranker in the intermediate denoising steps to inject sequence-level correlation. Experimental results demonstrate that EDLM outperforms existing diffusion models and achieves sampling speedup without sacrificing generation quality.

**Strengths:**

- The writing is clear, making the paper easily comprehensible.
- The idea to inject sequence-level correlation in the intermediate denoising steps is intuitive.

**Weaknesses:**

- The primary contribution lies in the energy-based reranker, which mostly follows the approach presented in https://arxiv.org/abs/2004.11714. In my view, I tend to have concerns about the novelty given previous work.
- The description of the motivation and the independent factorization in Equation 6 does not seem entirely accurate. For line 156 "$\mu_\theta$ predicts each token in x_0 independently", it may be more precise to state that the sampling operation over the output distribution is done independently, as during training, predictions for all masked token positions are made simultaneously, leading to their hidden representations being interrelated. This implies that predicting a particular token will implicitly involve using representations of other tokens, which makes the the independent factorization in Equation 6 not entirely accurate.
- This paper only evaluates perplexity, and it remains uncertain whether such energy-based reranking translates effectively to downstream tasks.
- I tend to have concerns about the inference cost linked to repetitive sampling and energy function computations in each denoising steps.

**Questions:**

- What are the used window size and sampling size in Figure 1(a). I am somewhat confused about the 1.3x sampling speedup as EDLM-AR/NCE requires more computation on the intermediate steps. Could you please elaborate on the rationale behind the increased efficiency of EDLM-AR/NCE compared to EDLM?
- In the case of EDLM-NCE, it seems that when the value of t is large, the quality of multiple sampled $x_0$ might all degrade, leading to general closer low scores assigned by the classifier. Given this, why does adding the reranker in the early stages yield better results instead?
- In the context of EDLM-NCE, do you employ separate models or do you utilize a shared rerank and generator approach?
- I am somewhat confused about Figure 3. In my view, as the window size increases, one would expect a higher time cost. However, the graph appears to show the opposite trend. Could you provide insight into this discrepancy?

---

> ### Author Response · Authors · 2024-11-20
> **Author Response Part 1**
>
> Thank you for your constructive feedback and questions! The responses to your questions are listed below, and paper is updated with updates colored in purple:
>
> > **[W1] Relevance to previous work on EBM for autoregressive models.**
>
> Yes, [1] studied similar accumulated error problems (*a.k.a* exposure bias) in the autoregressive model setting and also proposed residual energy-based models for modeling global context. However, their methodology and analysis primarily focus on improving autoregressive models. By contrast, we study how to combine energy-based models with diffusion models, which is fundamentally different from theirs. Particular differences include:
> 1. Our Mask Diffusion EBM formulation inspires the idea of leveraging pretrained AR as EBM, while [1] can only on NCE training.
> 2. Even in NCE training, our framework involves new training objectives where positive and negative sampling distributions are from true and learned diffusion posteriors.
> 3. We introduce hybrid sampling combining both the denoising process and importance sampling.
> 4. We introduce the estimator of the variational lower bound of diffusion likelihood for evaluating the perplexity.
>
> But following your suggestion, we realized we lack sufficient mention of this relevant work, and we have incorporated the content into related work.
>
> [1] Deng, Yuntian, et al. "Residual energy-based models for text generation." arXiv preprint arXiv:2004.11714 (2020).
>
> > **[W2] The description of the independent factorization in Equation 6.**
>
> In the paper, we use independent factorization in Equation 6 to represent that the distribution over the output distribution is factorized as a product of independent distributions where sampling is conducted independently.
>
> Although the hidden representation is shared, this can be viewed as using shared parameters $\theta$ to model the independent distributions. This cannot model the correlations, since in the input space there is no information on other missing tokens. This argument is also used across other literatures [2].
>
> [2] Lezama, Jose, et al. "Discrete predictor-corrector diffusion models for image synthesis." The Eleventh International Conference on Learning Representations. 2022.
>
> > **[W3] Effectiveness on downstream tasks.**
>
> We follow your suggestion and test our EDLM on an additional semi-autoregressive generation setting. We follow exactly the same setting as MDLM Sec 4.2 [3], but change the sampling from diffusion denoising to energy-based importance sampling. In short, the model aims to generate sequences with 2048 tokens by autoregressively generating 512-length blocks. The Gen. PPL results are shown as follows:
>
> | Method | SSD-LM | MDLM | EDLM |
> |-|-|-|-|
> | Gen. PPL |  35.43 | 27.18 | 22.62 |
>
> As shown in the semi-autoregressive results, we also notice similar effectiveness translates to this downstream generation task.
>
> [3] Sahoo, Subham Sekhar, et al. "Simple and Effective Masked Diffusion Language Models." arXiv preprint arXiv:2406.07524 (2024).
>
> > **[W4] Additional inference cost.**
>
> Yes, our method will require more inference cost due to additional computation for importance sampling. However, we want to highlight that our method follows the current trend of language model acceleration research such as speculative decoding [4], where we use more computation to trade for shorter sampling time. Furthermore, with enough GPU memory, these computations can be conducted efficiently in parallel.
>
> [4] Leviathan, Yaniv, Matan Kalman, and Yossi Matias. "Fast inference from transformers via speculative decoding, 2023." URL https://arxiv. org/abs/2211.17192 (2022).

---

> ### Author Response · Authors · 2024-11-20
> **Author Response Part 2**
>
> > **[Q1] The window size and sampling size in Figure 1(a). How EDLM achieves 1.3x sampling speedup.**
>
> The window size is $0.2$ and the sampling size is $2$. Note that, the importance sampling can be conducted in parallel in GPU.
>
> Then, just as a conceptual example, as shown in Fig (1.a), EDLM with 512 sampling steps can achieve similar results as MDLM with 1024 steps. Then, MDLM takes 1024 forward calls of the network, while EDLM takes 512*(1+0.2)≈615 calls. Since a forward pass of diffusion and energy function share a similar network and will take a similar time, the speed up will be around 1024/612=1.67. But note that this is just a conceptual example, and in practice, we get the 1.3x speedup.
>
> > **[Q2] Why does adding the reranker in the early stages yield better results instead?**
>
> We think the reason is the model tends to make more errors at the start due to limited context and importance sampling will correct them. In the late stage, even if the energy can differentiate samples, the energy cannot make a difference there since all samples are bad.
>
> Besides, we would like to mention that Figure 2 visualizes positive and negative samples from true $x_t$, e.g., $x_t$ is generated by diffusing true $x_0$ instead of by sampling. Therefore, the energy behavior is not directly the phenomenon during sampling.
>
> > **[Q3] Parameterization of EDLM-NCE energy function.**
>
> It’s a separate model, though fine-tuned from the pre-trained diffusion model.
>
> > **[Q4] Discrepancy about Fig 3.**
>
> Thank you for your detailed check!! We mislabeled the window sizes reversely, and it is even different from our own Fig 1. (We labeled the importance sampling ending timestep, and therefore the window should be $1-$ this value). We have updated the figure in our new version.
>
> ------
>
> We hope our response could address your questions!

---

### Official Review · Reviewer_9hNB · 2024-10-29

**Soundness:** 3
**Presentation:** 3
**Contribution:** 3
**Rating:** 6
**Confidence:** 3

**Summary:**

This paper proposes using an autoregressive language model to help the sampling process of a Masked Discrete Diffusion Language Model (which is technically equivalent to a Masked Language Model) by rescoring a selected subset of the generated outputs. The authors further propose to train this model with noise contrastive estimation and find a method to estimate its PPL.

**Strengths:**

1. Using an AR to help the sampling of Masked Discrete Diffusion Language Model is natural and straightforward.
2. The experiments demonstrate good improvements compared with MDLM baseline.

**Weaknesses:**

**My main concern is that the core technique introduced in this work has been present in the literature for a long time.**

1. First, the framework of the absorbing discrete diffusion model is essentially the same as the BERT-like masked language model (MLM). The forward process corresponds to masking tokens in the input, while the backward process corresponds to predicting and remasking tokens during iterative generation in MLMs. Therefore, this work primarily explores how to apply an energy-based framework to assist MLMs in generating text.

2. Importantly, the core part of the proposed method is using an autoregressive model as an energy function to guide the sampling process of an MLM (referred to as the denoising model in this paper). **The introduced energy function ($−logp_{AR}(x_0) + logp_θ(x_0|x_t)$) and the Importance Sampling method (Algorithm 1) are identical to reranking (or noisy parallel decoding) methods used in MLM generation/non-autoregressive generation, which have long been established in prior literature.** Examples can be found in [1,2,3]. Actually, the practice of using an AR model to rescore multiple outputs from an MLM has long been a standard baseline in non-autoregressive generation tasks.

[1] Non-autoregressive Neural Machine Translation, 2017
[2] Mask-Predict: Parallel Decoding of Conditional Masked Language Models, 2019
[3] Fully Non-autoregressive Neural Machine Translation: Tricks of the Trade, 2020

**Questions:**

Section 4.3 is not very clear to me. Could you provide a detailed explanation of how you estimated PPL?

---

> ### Author Response · Authors · 2024-11-20
>
> Thank you for your constructive feedback and questions! The responses to your questions are listed below, and paper is updated with updates colored in purple:
>
> > **[W1] This work primarily explores how to apply an energy-based framework to assist MLMs in generating text.**
>
> We agree with your takeaway that our method can be summarized as combining EBM and MLM through the diffusion framework for text generation. However, we highlight the key technical challenges and innovations as follows:
> 1. Our Mask Diffusion EBM framework inspires training with different noise levels determined by diffusion schedule, which differs from BERT.
> 2. We introduce NCE training for the EBM, which involves new training objectives where positive and negative sampling distributions are from true and learned diffusion posteriors.
> 3. We develop hybrid sampling combining both the denoising process and importance sampling.
> 4. We introduce the estimator of the variational lower bound of diffusion likelihood for evaluating the perplexity.
>
> > **[W2] Using an AR model to rescore multiple outputs from an MLM is a standard baseline in non-autoregressive generation tasks**
>
> This is a good point. Though they also use the AR model, these methods typically use the AR to conduct ranking and selection to decode the most likely sample, where sampling methods are biased and cannot generate sampling according to learned distribution. By contrast, our EBM method can reduce sampling error and still induce unbiased sampling.
>
> Theoretically, according to the formulation of mask diffusion models (Equation 5), the unmasking and masking schedules are defined in closed form, and the heuristic sampling methods such as ranking and selecting will result in biased sampling results. As a result, even if these methods can generate high-quality samples, they may result in other issues such as low diversity. As we can see, the relevant works primarily focus on conditional generation such as translation, where the metrics are mainly about quality instead of distribution modeling.
>
> Empirically, furthermore, we also included SUNDAE [2] as the representation of these models in comparison, which is one of the SOTA models for translation. As shown in Tab 3 in the paper, SUNDAE achieves good results with an extremely small number of timesteps, but the diversity is obviously much worse than EBM methods. This indicates the biased sampling issue from these methods.
>
> [1] Non-autoregressive Neural Machine Translation, 2017
>
>
> [2] Savinov, Nikolay, et al. "Step-unrolled denoising autoencoders for text generation." arXiv preprint arXiv:2112.06749 (2021).
>
> > **[Q1] Explanation of how to estimate PPL**
>
> Since perplexity is calculated from likelihood, we will mainly explain the details of calculating the likelihood. The exact formulation is in Eq. 13. Given data $x_0$, for every timestep $t$ we will perturb it to $x_t$ and compute the denoising loss (term 1 in Eq 13). For the energy term, we will first calculate the energy of $(x_0, x_t)$ pair (term 2); besides, we will sample several generated $x_0$ by denoising $x_t$ using the diffusion model and compute the **average** energy (this **averaging** is just an intuitive explanation for Theorem 1) of those samples to approximate the partition function (term 3). All three are combined together to compute Eq 13 weighted by the weights $\frac{\alpha_t’}{1-\alpha_t}$.
>
> ------
>
> We hope our response could address your questions!

---

> > ### Comment · Reviewer_9hNB · 2024-11-26
> >
> > Thank you for your efforts. I have checked your response carefully and decided to increase my score.

---

> ### Author Response · Authors · 2024-11-24
> **Additional quick message**
>
> Dear reviewer,
>
> This is a quick message to let you know that we are working on incorporating the suggested related work on Non-autoregressive Neural Machine Translation into the paper. We are still polishing the content and rearranging the pages, and will post here once we finished adding the content. We expect to finish it by tomorrow.

---

> ### Author Response · Authors · 2024-11-25
>
> Thanks again for your feedback! We have updated the paper accordingly. Please let us know if you have other questions!

---

### Official Review · Reviewer_r3kY · 2024-11-02

**Soundness:** 4
**Presentation:** 4
**Contribution:** 2
**Rating:** 8
**Confidence:** 4

**Summary:**

This work presents a new discrete diffusion language model that uses an energy-based model (EBM) to introduce token dependencies within the text sequence, instead of factorizing the denoising distribution over individual tokens. In practice, this EBM is parameterized in a residual form, with the energy function implemented by either a pretrained auto-regressive Transformer or a fine-tuned bidirectional Transformer. To facilitate efficient sampling, the paper employs self-normalizing importance sampling and draws several samples followed by resampling based on energies to complete each denoising step. The resulting framework, called Energy-based Diffusion Language Models (EDLMs), demonstrates significant advancements over previous discrete diffusion baselines and achieves comparable or better performance to traditional auto-regressive language models on text generation.

**Strengths:**

- This paper is well-written and well-organized, presenting a simple yet elegant combination of energy-based and discrete diffusion models. EDLM demonstrates strong empirical results that match auto-regressive models with great improvements in sampling speed.
- The adaptation of auto-regressive models to a joint denoising distribution with masked inputs is innovative.
- The application and detailed analysis of importance sampling windows effectively improve the early sampling phases in discrete diffusion models, making a nice empirical contribution to the advancements of discrete diffusion language models.

**Weaknesses:**

- While the study aims to address the independence assumptions in discrete diffusion models through EBMs, there is insufficient examination of relevant prior research also involving EBMs for language modeling (e.g., [1]). Given the similarity between EDLMs and [1], a more detailed discussion and comparison would clarify the position and relevance of this study.
- Another concern lies in the significance of applying EBMs to discrete diffusion models. Although vanilla discrete diffusion processes do factorize the denoising distribution over tokens, the bidirectional attention mechanism in transformers can already capture dependencies among tokens within a sequence. Therefore, to account for errors due to parallel decoding, feeding the decoded sequence back into the transformer (i.e., the next denoising step) could potentially identify erroneous tokens and assign low likelihoods to those positions. There is extensive literature on filtering and remasking tokens at each denoising step [2, 3, 4]. An in-depth discussion highlighting the advantages of using an EBM to capture token dependencies would strengthen the paper’s argument.

[1] Deng, Yuntian, et al. "Residual energy-based models for text generation." *arXiv preprint arXiv:2004.11714* (2020).

[2] Ghazvininejad, Marjan, et al. "Mask-predict: Parallel decoding of conditional masked language models." *arXiv preprint arXiv:1904.09324* (2019).

[3] Savinov, Nikolay, et al. "Step-unrolled denoising autoencoders for text generation." *arXiv preprint arXiv:2112.06749* (2021).

[4] Zheng, Lin, et al. "A reparameterized discrete diffusion model for text generation." *arXiv preprint arXiv:2302.05737* (2023).

**Questions:**

1. For EDLM-AR and EDLM-coAR, which pretrained AR model is used and what is the parameter size of the model?
2. For all EDLM variants, a separate transformer model is used to calculate the energy, which effectively doubles the parameter count and might contribute significantly to performance gains. While the reported performance and efficiency improvements likely go beyond what would be expected from simply doubling the model size, it would strengthen the validity of these findings to include an empirical comparison or an in-depth discussion with a baseline with an equivalent parameter count.

---

> ### Author Response · Authors · 2024-11-20
>
> Thank you for your constructive feedback and questions! The responses to your questions are listed below, and paper is updated with updates colored in purple:
>
> > **[W1] Relevance to previous work on EBM for autoregressive models.**
>
> Yes, [1] studied similar accumulated error problems (*a.k.a* exposure bias) in the autoregressive model setting and also proposed residual energy-based models for modeling global context. However, their methodology and analysis primarily focus on improving autoregressive models. By contrast, we study how to combine energy-based models with diffusion models, which is fundamentally different from theirs. Particular differences include:
> 1. Our Mask Diffusion EBM formulation inspires the idea of leveraging pretrained AR as EBM, while [1] can only rely on NCE training.
> 2. Even in NCE training, our framework involves new training objectives where positive and negative sampling distributions are from true and learned diffusion posteriors.
> 3. We introduce hybrid sampling combining both the denoising process and importance sampling.
> 4. We introduce the estimator of the variational lower bound of diffusion likelihood for evaluating the perplexity.
>
> But following your suggestion, we realized we lack sufficient mention of this relevant work, and we have incorporated the content into related work.
>
> [1] Deng, Yuntian, et al. "Residual energy-based models for text generation." arXiv preprint arXiv:2004.11714 (2020).
>
> > **[W2] Discussion about relevant methods for filtering and remasking tokens at each denoising step.**
>
> This is a good point. We agree that diffusion provides likelihoods for all tokens and we can use this information to potentially identify the most incorrect tokens, but these sampling methods are biased and cannot generate sampling according to learned distribution. By contrast, our EBM method can reduce sampling error and still induce unbiased sampling.
>
> Theoretically, according to the formulation of mask diffusion models (Equation 5), the unmasking and masking schedules are defined in closed form, and the heuristic sampling methods such as deterministic argmax decoding [2] will result in biased sampling results. As a result, even if these methods can generate high-quality samples, they result in other issues such as low diversity. As we can see, the relevant works [2-4] primarily focus on conditional generation such as translation or paraphrasing, where the metrics are mainly about quality instead of distribution modeling.
>
> Empirically, furthermore, we also included SUNDAE [3] as the representation of these models in our benchmark comparison. As shown in Tab 3 in the paper, SUNDAE achieves good results with an extremely small number of timesteps, but the diversity is obviously much worse than EBM methods. This indicates the highly biased sampling issue from these methods.
>
> [2] Ghazvininejad, Marjan, et al. "Mask-predict: Parallel decoding of conditional masked language models." arXiv preprint arXiv:1904.09324 (2019).
>
>
> [3] Savinov, Nikolay, et al. "Step-unrolled denoising autoencoders for text generation." arXiv preprint arXiv:2112.06749 (2021).
>
>
> [4] Zheng, Lin, et al. "A reparameterized discrete diffusion model for text generation." arXiv preprint arXiv:2302.05737 (2023).
>
> > **[Q1] Details of pretrained AR model.**
>
> It’s a GPT-2 small model also trained on OpenWebText data, which uses the same model scale and training data as diffusion models. It’s a 12-layer transformer with 768-dimensional embeddings and 12 attention heads, resulting in 124M parameters.
>
> > **[Q2] Baseline with an equivalent parameter count.**
>
> This is a good point. EDLM is composed of a diffusion model and an energy-based model, which introduces $2\times$ number of parameters. To clearly show that the performance improvements come from the framework design instead of model size, we follow your insight and conduct another ablation setting, where we double the number of layers and train a new MDLM for ~7 days. The key results on PPL are shown below:
>
> | OpenWebText | PTB | Wikitext | LM1B | Lambada | AG News | Pubmed | Arxiv |
> | - | - | - | - | - | - | - | - |
> | 23.77 | 96.10 | 32.83 | 66.88 | 46.98 | 61.20 | 41.11 | 36.54 |
>
> As shown in the results, we didn’t notice a significant improvement by simply doubling the model size. This yields a more fair comparison to suggest the effectiveness of our proposed EDLM framework.
>
> ------
>
> We hope our response could address your questions!

---

> > ### Comment · Reviewer_r3kY · 2024-11-24
> >
> > I appreciate the authors' comprehensive responses and efforts to address my concerns. I agree with their points that
> >
> > - Reranking and remasking through the model itself with likelihood heuristics would reduce sampling fidelity and output diversity.
> > - It is interesting that even when the parameter count of MDLM is doubled, the performance remains similar to the baseline on these tasks. This observation suggests that either perplexity tasks might be saturated or that substantial challenges exist in scaling diffusion models. Nevertheless, I believe the benefits of introducing the EBM should surpass those of simply increasing the parameter count.
> >
> > After carefully reviewing the additional information, I have increased my score to 8 to reflect the paper's contributions. However, upon reading the other reviewers' comments, I concur that this work should discuss its connections to [1] and the non-autoregressive generation literature (e.g. [5]) in greater depth (e.g. elaborate on the updated Lines 101–103). Furthermore, the analyses concerning the reranking and/or remasking literature [2-4] should be incorporated into the main manuscript, particularly the distribution modeling and diversity aspects, which would further enhance the manuscript.
> >
> > [1] Deng, Yuntian, et al. "Residual energy-based models for text generation." *arXiv preprint arXiv:2004.11714* (2020).
> >
> > [2] Ghazvininejad, Marjan, et al. "Mask-predict: Parallel decoding of conditional masked language models." *arXiv preprint arXiv:1904.09324* (2019).
> >
> > [3] Savinov, Nikolay, et al. "Step-unrolled denoising autoencoders for text generation." *arXiv preprint arXiv:2112.06749* (2021).
> >
> > [4] Zheng, Lin, et al. "A reparameterized discrete diffusion model for text generation." *arXiv preprint arXiv:2302.05737* (2023).
> >
> > [5] Gu, Jiatao, and Xiang Kong. "Fully non-autoregressive neural machine translation: Tricks of the trade." *arXiv preprint arXiv:2012.15833* (2020).

---

> > > ### Author Response · Authors · 2024-11-24
> > >
> > > Thank you very much for your feedback and recognition of our efforts! We greatly appreciate your suggestions!
> > >
> > > This is a quick reply to confirm that we received your feedback, and let you know that we are working on incorporating the discussions into the paper and rearranging the pages. We will post here once we finished adding the content, and we expect to finish it by tomorrow.

---

> > > ### Author Response · Authors · 2024-11-25
> > >
> > > Thanks again for your feedback! We have updated the paper accordingly. Please let us know if you have other questions!

---

### Official Review · Reviewer_LAtw · 2024-11-02

**Soundness:** 2
**Presentation:** 3
**Contribution:** 3
**Rating:** 8
**Confidence:** 2

**Summary:**

This paper presents a new Discrete Diffusion Model that models an energy function $E_\phi$ to improve sampling procedure from an existing Diffusion Model. The authors propose a novel sampling procedure that incorporates importance sampling steps during generation.

**Strengths:**

- The paper is clearly written and easy to follow.
- The method is intuitive and based on a solid mathematical foundation.
- The experiments follow the standard setup for evaluating Diffusion Language Models, making it easy to compare with other methods.

**Weaknesses:**

- This method requires a pre-trained discrete diffusion model, which increases the overall computational requirements. Thus, it may be unfair to compare it directly with simpler methods like MLDM.
- While the proposed method reduces the Gen PPL metric, it also decreases the entropy of generated texts. One could even argue that it produces similar results to MLDM in terms of Gen PPL and entropy.

**Recommended Experiments**:

It would be interesting to see a more detailed trade-off between entropy and Gen PPL compared to the baselines and the proposed model. Although Figure 1 provides some insight, a separate figure with Entropy/Gen PPL axis would address concerns about the notably reduced entropy.

As mentioned, diffusion models are not as computationally efficient as simpler AR models. However, it may be possible to improve efficiency by allocating different computational budgets for training the base $p_\theta$ and $E_\phi$ models. Given the potential compute intensity of this experiment, a simpler approach might involve training the existing MLDM for the additional steps used to train the energy function  $E_\phi$ , then reporting the evaluation results.

**Questions:**

- Is it correct that the number of parameters used for generation is approximately doubled? While generation time is important, it would also be useful to know the VRAM requirements for generation.
- Is it possible to apply the importance sampling steps at a different interval, i.e. at the middle of the generation rather than at the start? This adjustment might yield improved performance.

---

> ### Author Response · Authors · 2024-11-20
>
> Thank you for your constructive feedback and questions! The responses to your questions are listed below, and paper is updated with updates colored in purple:
>
> > **[W1] Unfair to compare with MDLM.**
>
> This is a good point. EDLM is composed of a diffusion model and an energy-based model, which introduces $2\times$ number of parameters. To clearly show that the performance improvements come from the framework design instead of model size, we follow your insight and conduct another ablation setting, where we double the number of layers and train a new MDLM for ~7 days. The key results on PPL are shown below:
>
> | OpenWebText | PTB | Wikitext | LM1B | Lambada | AG News | Pubmed | Arxiv |
> | - | - | - | - | - | - | - | - |
> | 23.77 | 96.10 | 32.83 | 66.88 | 46.98 | 61.20 | 41.11 | 36.54 |
>
> As shown in the results, we didn’t notice a significant improvement by simply doubling the model size. While the neural network may become more expressive, the original model cannot account for correlations regardless of model size, which may explain the lack of a significant improvement. This yields a more fair comparison to suggest the effectiveness of our proposed EDLM framework.
>
> > **[W2] GenPPL vs diversity trade-off is similar to MDLM.**
>
> We actually agree with your observation that the GenPPL vs diversity is similar. Note that, as in Fig1, MDLM’s diversity will also decrease with increased sample steps. EDLM aims to reduce the sampling errors and therefore has a similar performance as MDLM but with much fewer sampling timesteps.
>
> > **[R1] More detailed trade-off between entropy and GenPPL.**
>
> Thank you for your suggestion and we prepared a figure for this comparison. We added it in Fig 4 in the updated paper. The figure supports our augment above, where we actually show a tradeoff similar to MDLM but with reduced time steps.
>
> > **[R2] Improve efficiency for training the energy function $\phi$.**
>
> This is a good point and actually, in the EDLM-NCE setting, we train the energy $\phi$ by finetuning from the pretrained diffusion model $\theta$. We mention this information in Sec 5.1 **Implementation Details** paragraph: *we finetune the pretrained MDLM, with the energy function computed by projecting the mean-pooled last token embeddings down to a single scalar value*.
>
> Besides, we would also like to highlight that in the other $\phi$ implementation EDLM-AR, we can leverage pretrained AR LLM and avoid additional training costs, which is even more efficient.
>
>
> > **[Q1] Memory requirements for generation.**
>
> Yes, we approximately double the number of parameters. In FP-32 precision, the VRAM for MDLM is around 500MB while EDLM is 1GB. However, we emphasize that the VRAM during inference mainly comes from activation instead of the model itself, so this is not a bottleneck of EDLM.
>
> > **[Q2] Importance sampling at different intervals.**
>
> Yes we can use different intervals. But as noted in Sec 4.4 **Sampling window** paragraph, we notice that with the same window, applying it on early stage sampling would always have better performance. For example, with 1024 sampling steps and a fixed window of 0.2, when applying it at different intervals we have the following results:
>
> |Time Interval | 0.0-0.2 | 0.2-0.4 | 0.4-0.6 | 0.6-0.8 | 0.8-1.0 |
> |-|-|-|-|-|-|
> | GenPPL | 33.43 | 32.49 | 31.32 | 31.15  | 30.11 |
>
> which shows that applying it at the start still shows the best result. We think the reason is the model tends to make more errors at the start due to limited context and importance sampling will correct them. Therefore we focus on discussing this case in the paper.
>
> ------
>
> We hope our response could address your questions!

---

> > ### Comment · Reviewer_LAtw · 2024-11-23
> >
> > Thank you for the insightful answers. The experiments you provided addressed all of my concerns and I will improve my score. I do have a few minor questions about the results:
> > > we double the number of layers and train a new MDLM for ~7 days
> >
> >
> > What if we double the amount of computation in training steps, rather than in model size? Additionally, I believe that doubling the number of layers is not the optimal scaling for the given number of parameters. Could the authors elaborate on that? From my perspective, the results would not change significantly from the ones you provide, but it would be useful to see discussions on this in future revisions.
> > > For example, with 1024 sampling steps and a fixed window of 0.2, when applying it at different intervals we have the following results...
> >
> >
> > How does entropy change in this setup? When considering diffusion LMs, I believe it is important to consider two metrics.

---

> > > ### Author Response · Authors · 2024-11-24
> > >
> > > Thank you very much for your feedback and recognition of our efforts! We appreciate your suggestions and will incorporate the discussions into the final version. Please let us know if you have any other questions affecting your confidence!
> > >
> > > For your follow-up questions:
> > >
> > > > What if we double the amount of computation in training steps, rather than in model size?
> > >
> > > This is a good point. Given the short response time window, we can only try doubling the model size instead of doubling the training time. However, I would suppose the results will be similar since we did try continuing training of the MDLM checkpoint but see no improvement anymore, which indicates the current training length is already a converged model at this size. We're conducting formal double-time training now and will incorporate relevant results into the final version.
> > >
> > > > How does entropy change in this varying window
> > >
> > > The diversities don't have a very different value, around 7.3. The results for both metrics are:
> > >
> > > |Time Interval | 0.0-0.2 | 0.2-0.4 | 0.4-0.6 | 0.6-0.8 | 0.8-1.0 |
> > > |-|-|-|-|-| -|
> > > | GenPPL | 33.43 | 32.49 | 31.32 | 31.15  | 30.11 |
> > > | Diversity | 7.4 | 7.4 | 7.3 | 7.3  | 7.3 |
> > >
> > > As there is no huge difference in the generative perplexity, it's reasonable that the methods show similar diverities.

---

### Meta-Review · Area_Chair_5VcA · 2024-12-20

**Metareview:**

This paper introduces Energy-based Diffusion Language Models (EDLM), which addresses token dependencies in discrete diffusion models by incorporating an energy-based model for reranking during the denoising process. The paper's main strengths are its clear presentation, strong empirical results showing improved performance over baseline discrete diffusion models, and the introduction of an innovative sampling procedure that combines denoising with importance sampling. The key weaknesses include concerns about novelty given similarities to previous work on energy-based models for text generation and non-autoregressive translation, potential computational overhead from repeated sampling, and initially limited evaluation on downstream tasks. The paper is recommended for acceptance primarily due to its technical soundness, clear improvements over baselines, and practical significance in advancing discrete diffusion models for text generation, even though some reviewers initially had concerns about novelty.

**Additional Comments On Reviewer Discussion:**

During the discussion phase, reviewers raised several important points regarding model comparison fairness, connections to prior work, downstream task performance, and computational requirements. The authors partially addressed these concerns by: (1) conducting new experiments with doubled model size to demonstrate improvements come from the framework rather than parameter count, (2) adding detailed comparisons to prior work on energy-based models and non-autoregressive generation, (3) providing additional downstream task results showing improved performance in semi-autoregressive generation, and (4) clarifying technical details about sampling efficiency and implementation. While the ablation study with doubled model size helped demonstrate that the improvements aren't solely due to increased parameters, the fundamental requirement of an external model (either pretrained AR or fine-tuned from diffusion, which uses massive compute) remains a limitation that requires additional computational resources and makes deployment more complex compared to standalone approaches. After the discussion, two reviewers increased their scores, while one maintained their score. Though some concerns about computational overhead, reliance on external model, and model complexity remain, the demonstrated improvements in generation quality and the authors' thorough theoretical framework make this paper a worhtwhile contribution to the community, albeit with noted limitations.

---

### Decision · Program_Chairs · 2025-01-22

Accept (Poster)